# Trajectory-Level Data Augmentation for Offline Reinforcement Learning

**Tobias Schmähling** [1]   **Matthias Burkhardt** [1]   **Tobias Windisch** [1]

## Abstract

We propose a data augmentation method for offline reinforcement learning, motivated by active positioning problems. Particularly, our approach enables the training of off-policy models from a limited number of suboptimal trajectories. We introduce a trajectory-based augmentation technique that exploits task structure and the geometric relationship between rewards, value functions, and mathematical properties of logging policies. During data collection, our augmentation supports suboptimal logging policies, leading to higher data quality and improved offline reinforcement learning performance. We provide theoretical justification for these strategies and validate them empirically across positioning tasks of varying dimensionality and under partial observability.

## 1. Introduction

Offline reinforcement learning promises to learn effective decision-making policies from static, pre-collected datasets, avoiding the cost and risk of online exploration (Levine et al., 2020). This is particularly attractive in real systems, where trial-and-error interaction is expensive or unsafe. Yet the central challenge of offline RL is equally well known: because learning is constrained to the support of the dataset, distribution shift between the learned policy and the data-generating behavior can lead to severe extrapolation errors and brittle, suboptimal performance. Contemporary methods therefore rely on conservative updates, regularizing toward the behavior distribution or warm-starting from the logging policy before attempting improvement. However, these algorithmic safeguards do not remove the core dependency on the data itself. Consequently, offline RL performance can depend strongly on the quality of the logging policy that produced the data. Prior evidence shows that dataset selection can outweigh algorithmic differences (Schweighofer

et al., 2022; Fu et al., 2021; Yarats et al., 2022), suggesting that the logging policy effectively sets the attainable frontier for an offline learner. While the field has developed a rich set of algorithms for coping with imperfect data (see Section 1.2), actionable principles for improving the data-generating process remain scarce.

We observe an algorithmic gap in what can be done with a given logging policy to improve RL, with pure offline learning on the one end and offline-to-online fine-tuning on the other. Motivated by this, we seek to understand ways in the middle, particularly how logging policies can be augmented already *during collection* in a principled way to generate better data for RL. While prior work has studied the effect of exploration (Zhang et al., 2023), we focus on exploiting logged data to improve learning. Here, a key practical obstacle to improve datasets that is easily overlooked is the *hand-off problem*. In many applications, the logging policy is not a stochastic, exploratory controller, but a deterministic, scripted process with internal state. Injecting a better action mid-trajectory can invalidate its assumptions, forcing a restart before execution can safely resume.

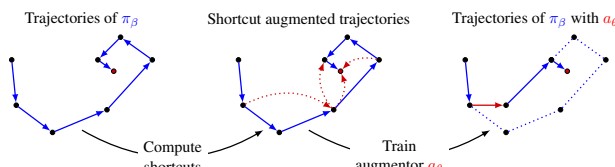

Figure 1. Overview of LIFT.

In this paper, we study logging policy augmentation in the context of *active positioning problems* that capture both partial observability and fine tolerance demands that make online RL particularly costly, while also reflecting the prevalence of deterministic procedures in practice, making them an ideal testbed for offline RL in general and logging augmentations in particular. Additionally, their contextual and geometric structure enables a theoretically grounded analysis of when and why augmentations are beneficial. They require placing an object precisely at a desired position by an end-effector, spanning a wide range of challenging RL problems, from high-precision positioning tasks as alignments of lens systems (Burkhardt et al., 2025) and camera and telescope assembly (Bräuniger et al., 2014; Upton et al., 2006), the alignments of laser optics (Rakhmatulin et al., 2024;

[1]University of Applied Sciences Kempten, Kempten, Germany. Correspondence to: Tobias Schmähling <tobias.schmaehling@hs-kempten.de>.

*Proceedings of the 43$^{rd}$ International Conference on Machine Learning*, Seoul, South Korea. PMLR 306, 2026. Copyright 2026 by the author(s).

Sorokin et al., 2020), to robot manipulation tasks (Plappert et al., 2018).

## 1.1. Contributions

We introduce *LIFT*, short for logging improvement via fine-tuned trajectories, a framework that enhances punctual data collection for offline RL. Specifically, we propose a novel augmentation scheme (Section 4) that keeps the logging policy in control while enabling optimistic probing by an *augmentor* trained while data is collected. The augmentor's goal is to skip redundant and unnecessary sub-trajectories during collection and to smooth hand-offs between itself and the logging policy. A key challenge here is that the augmentor has to suggest beneficial actions while being trained with very limited data. A central innovation of our work is to leverage the geometric structure of the logged trajectories to identify *shortcuts*, that is, actions that point towards states with higher value. Identifying shortcuts is non-trivial in general due to distortions in the dynamics and the partial observability. We prove in Section 3 under which conditions such shortcuts can be reliably identified in logged data, and we devise an algorithm to extract them from this data (Algorithm 1). Finally, Section 5 presents a systematic study that underlines the strength and generality of our approach by analyzing the effect of the logging policy, transition behavior, dimensionality, and informativeness of observations on policy performance across a diverse class of active positioning tasks. We implemented the shortcut augmentation in d3rlpy (Seno & Imai, 2022), following its transition picker protocol, which allows our static augmentation method to be integrated into any RL algorithm implemented in d3rlpy by adding a single line of code. The source code and integration examples are available on GitHub.[1]

## 1.2. Related Work

A central challenge in offline RL is overestimating values for out-of-distribution actions. Methods address this either by constraining the learned policy toward the logging distribution or b learning pessimistic value functions. Representative approaches include behavior regularization via BC losses or divergence penalties (Fujimoto et al., 2019; Fujimoto & Gu, 2021; Tarasov et al., 2023), pessimistic critics (Kumar et al., 2020), or expectile-based policy extraction (Kostrikov et al., 2022). Methods depending on regularizations are sensitive to hyperparameters and they often limit the policy to stay close to the behavior, for instance due to safety constraints, which can be detrimental if the behavior is highly suboptimal. Moreover, several studies note that algorithm performance is highly sensitive to dataset composition (Fu et al., 2021; Hong et al., 2023), that is, mixing suboptimal trajectories with expert data. Prior work has

studied intensively the importance of high-coverage (Yarats et al., 2022; Wagenmaker et al., 2025) and expertness of datasets (Kumar et al., 2022; Corrado et al., 2024) for offline RL. This has been underpinned by the investigations in (Schweighofer et al., 2022), where scores are designed that measure exploitation and exploration capabilities of datasets and how these affect algorithmic performance of offline RL methods. While Ghugare et al. (2024) discuss limitations in combinatorial generalization ('stitching') from an algorithmic perspective, our work addresses a complementary problem at the data level through trajectory-level augmentation. Increasing the dataset diversity via data augmentations is another line of work to mitigate narrow data distributions. In (Andrychowicz et al., 2017), an augmentation scheme for sparse reward in robotic manipulation tasks is proposed that re-labels goals and states in logged trajectories to create additional successful transitions. Augmentations for problems with image observations have been studied extensively in the literature, where it was shown that rather simple image augmentations (Laskin et al., 2020; Sinha et al., 2022), such as random cropping, or utilizing causal techniques (Pitis et al., 2020) can significantly improve sample efficiency. Recently, diffusion-based techniques have been proposed that generate synthetic trajectories in order to make offline RL more robust (Li et al., 2024; Lee et al., 2024; Lu et al., 2023). In contrast to purely *offline* augmentations on static datasets, hybrid schemes that actively enhance data collection are more relevant to our work. A common hybrid approach warm-starts online reinforcement learning from an offline-trained policy and continues training with newly collected online data. Prior work shows that, combined with careful sampling schemes and network architectures (Ball et al., 2023) or policy regularization (Nair et al., 2018), this can yield strong initializers for online learning. Nevertheless, these methods still require rather long online fine-tuning or high-quality offline datasets, neither of which is typically available in active positioning tasks. A more subtle scheme is to let an expert guide the data collection process, like in GuDA (Corrado et al., 2024), where human-guidance is interleaved to direct trajectories toward success. Another relevant line of work is to weave online transitions into logging policies as in iterative offline RL (IORL) (Zhang et al., 2023). Here, exploratory actions are injected to discover unexplored regions in state-action space while training an offline RL agent on the generated trajectories. This approach is discussed in Section 4. Our approach is similar in spirit, but instead of exploring we want to exploit shortcuts in the trajectories to make hand-offs seamless and effective.

## 2. Active Positioning

In this section, we introduce the specific framework for active positioning problems building upon framework for

---

[1] https://github.com/HS-Kempten/lift

active alignments introduced in (Burkhardt et al., 2025). There, active positioning problems are modelled as an *episodic* and *contextual* POMDP (Modi et al., 2018). Specifically, the state is decomposed in the current position $s \in \mathcal{P}$ with $\mathcal{P}$ a bounded subset of $\mathbb{R}^m$ and a static context parameter $W \in \mathcal{W}$, that is $\mathcal{S} = \mathcal{P} \times \mathcal{W}$. The actions can be selected from a subset $\mathcal{A}$ of $\mathbb{R}^d$. Applying an action $a \in \mathcal{A}$ at state $(s, W)$ gives the new state $(s', W)$ with $s' = f(s, a, W)$, where $f : \mathcal{P} \times \mathcal{A} \times \mathcal{W} \to \mathbb{R}^d$ is a parametrized *distortion function*. Throughout we assume that $f(s, 0, W) = s$. Our running example is $f(s, a, W) = s + W \cdot a$ with $W \in \mathbb{R}^{d \times d}$, but we also consider non-linear and non-continuous distortions. Importantly, as $W$ stays constant throughout each episode, so is the extent of the distortion. One can think of $W$ as variances introduced by the gripping of an object, variances within an object, or conditions of the goal to be reached. In robotic arm positioning, for instance, $W$ can model the imprecision of the end-effector due to load or joint friction as well as where the target $s_W$ is located.

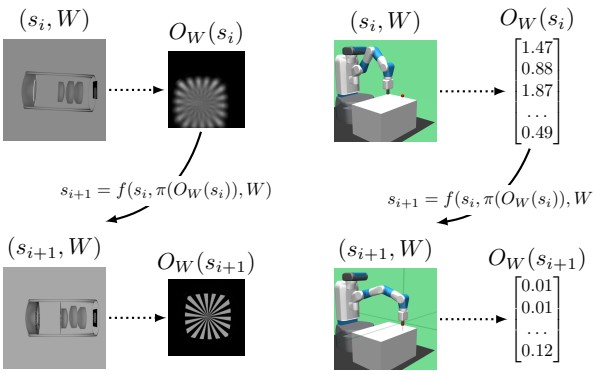

*Figure 2.* Active positioning of a lens systems (Burkhardt et al., 2025) (left) and an end-effector (Plappert et al., 2018) (right).

In each episode, the goal is to navigate from a random initial position $s_0$ and randomized context $W$ to a terminal state $s_W \in \mathbb{R}^d$. The reward observed when applying $a$ at $(s, W)$ is $R(s, a, W) = -\|f(s, a, W) - s_W\|$, i.e. the negative remaining distance to the terminal state. An episode ends once the state is sufficiently close to $s_W$ or an upper limit of steps is reached. Formally, the terminal states are all within the set $\{(s, W) \in S : \|s - s_W\| \leq \theta\}$. Typically, $W$ cannot be observed directly, often even $s$ cannot. Instead, an often high-dimensional and noised output $O(s, W) \in \mathcal{O}$ is observed, which is controlled by a conditional probability density function depending on $s$ and $W$. In robotic arm positioning, the observation can come from a camera mounted on the end-effector or from sensors measuring forces and torques. We call $(\mathcal{P}, \mathcal{W}, \mathcal{O}, f, \gamma)$ an *active positioning problem*. This framework covers various industrial use cases, from robot arm positioning, to active alignments of optical devices (Figure 2).

Although active positioning problems can also be considered as black-box optimization problems (Burkhardt et al., 2025), they are inherently RL problems where symmetries and ambiguities in the need to be actively explored. For instance, the observation space is typically highly symmetric and context-dependent: states $s$ and $s'$ that are far apart can yield very similar observations $O(s, W) \approx O(s', W)$, while the same state can produce very different observations $O(s, W)$ and $O(s, W')$ under different contexts. Additionally, safety constraints and physical limitations often restrict the action space $\mathcal{A}$ so that the optimal state cannot be reached in one step and a sequence of informed actions is required. In the RL formulation, a *policy* $\pi : \mathcal{A} \times \mathcal{O} \to \mathbb{R}$ is a mapping of observations and actions to likelihood and the dynamics of the combined system works as follows: At a given state $(s, W)$, $O(s, W)$ is observed, an action $a$ is sampled from $\pi(\cdot, O(s, W))$, and the system moves to the new state $s' = f(s, a, W)$. Note that $a$ and $s$ do not need to have same dimensionality. Starting from $(s_0, W) \in \mathcal{S}$, the dynamics yields a trajectory $(s_0, W), \ldots, (s_k, W)$. The goal is to find $\pi$ maximizing $J(\pi) := \mathbb{E}_{s_0, W} \left[ \sum_{i=0}^{k} -\gamma^i \|s_i - s_W\| \right]$, where $\gamma \in (0, 1)$ is a *discount factor*. Clearly, $J(\pi) = \mathbb{E}_{s_0, W}[V^\pi(s_0, W)] = \mathbb{E}_{s_0}[V^\pi(s_0)]$ with $V^\pi$ the state-value function and $V^\pi(s) := \mathbb{E}_{W \sim \mathcal{W}}[V^\pi(s, W)]$.

## 3. Theory of Shortcut Augmentations

In active positioning, good trajectories reach the optimal position in as few steps as possible. Although most logging policies used in applications visit states that are close to the optimal state, they often produce long and redundant trajectories. Our core idea is to train agents on synthetic trajectories distilled from these imperfect data, which are more direct and goal-reaching. Intuitively, we want the agent to *skip* parts of the trajectory that do not add much value — for example, going straight instead of replicating zig-zag movements or detours present in the logged data (Figure 1). However, improving logged trajectories is not straightforward. For instance, assume a collected trajectory of $\pi_\beta$ contains a sub-trajectory $(s_i, W), (s_{i+1}, W), \ldots, (s_j, W)$ with actions $a_i, \ldots, a_{j-1}$, representing a long detour, like a zig-zag movement, from $s_i$ to $s_j$. Clearly, going directly from $s_i$ to $s_j$ would yield a trajectory with higher return. However, naively applying the accumulated action $a = a_i + a_{i+1} + \ldots + a_{j-1}$ at $s_i$ will not necessarily land exactly at $s_j$ due to distortions in the dynamics induced by $f$. Even small misplacements, that is ending up close to $s_j$ but not exactly at $s_j$, can cause significant value degradation if the value function $V^{\pi_\beta}$ is not stable in the vicinity of $s_j$. Worse, applying $a$ at $s_i$ may even move us in the opposite direction, away from $s_j$, with no guarantee that the new state has a higher value than $s_i$. Here, the length of

the action $a$, the value gap between $s_i$ and $s_j$, the stability of $V^{\pi_\beta}$ around $s_j$, and the distortion in the dynamics at $s_i$ all play a role. In this section, we identify conditions under which the accumulated action $a$ is guaranteed to be beneficial. All proofs are in Section A. We call a policy $\pi$ *distance-improving* if for all $W \in \mathcal{W}$ we have for two subsequent states $(s_i, W)$ and $(s_j, W)$ with $i < j$ visited by the policy that $\|s_j - s_W\| < \|s_i - s_W\|$. In other words, the reward along a trajectory of $\pi$ is strictly increasing. We restrict to deterministic logging policies $\pi$, so that the contextual but deterministic dynamics given by $f$ implies that $V^\pi(s, W)$ is exactly the return of $\pi$ starting from $(s, W)$.

**Proposition 3.1.** *Let $\pi$ be distance-improving and $(s, W), (s', W) \in \mathcal{S}$ on a trajectory where $(s, W)$ is prior to $(s', W)$, then $\gamma V^\pi(s', W) - V^\pi(s, W) \geq \|s' - s_W\|$.*

Focusing on distance-improving logging policies allows us to formalize what it means for an action to be beneficial.

**Definition 3.2.** *Let $\pi$ be a policy, $(s, W) \in \mathcal{S}$ a state, and $a \in \mathcal{A}$ an action with $s' = f(s, a, W)$. If $\gamma V^\pi(s', W) - V^\pi(s, W) \geq \|s' - s_W\|$, then $a$ is a $\pi$-shortcut at $(s, W)$.*

Note that shortcuts depend on the latent information $W$, not $s$ alone. The remainder of this section studies how to find shortcuts in offline trajectories. To do so, consider a short trajectory $(s_0, W), (s_1, W), (s_2, W)$ from a distance-improving policy $\pi$ with actions $a_0$ and $a_1$ (Figure 3a). Clearly, any action $a$ with $s_2 = f(s_0, a, W)$ is a $\pi$-shortcut and thus beneficial. However, because of non-linearities in $f$, applying $a_0 + a_1$ at $s_0$ is not guaranteed to reach $s_2$. Hence, we must ensure that $a_0 + a_1$ leads near $s_2$ requiring to control the placement errors induced by $f$. For linear dynamics $f(s, a, W) = s + W \cdot a$ with $W \in \mathbb{R}^{m \times d}$, any accumulated action is a shortcut, irrespective of $V^\pi$:

**Proposition 3.3.** *Let $f(s, a, W) = s + W \cdot a$, $(s_i, W)$, $(s_j, W)$ with $i < j$ on a trajectory of a distance improving policy $\pi$ and $a_i, \ldots, a_{j-1}$ the actions $\pi$ applied to get from $s_i$ to $s_j$. Then $\sum_{k=i}^{j-1} a_k$ is a $\pi$-shortcut for $s_i$.*

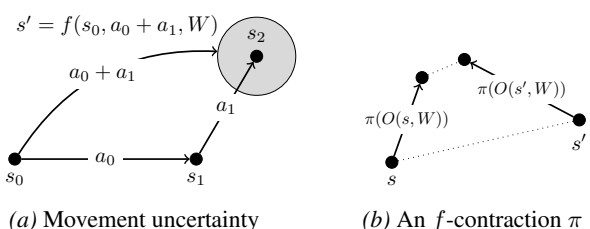

*(a)* Movement uncertainty        *(b)* An $f$-contraction $\pi$

*Figure 3.* Interactions of policy with movement dynamics.

Extending Proposition 3.3 to non-linear dynamics $f$ is not trivial. Generally, we want to have that accumulating actions along a trajectory does not lead to too much placement uncertainty, which is typically the case in real-world positioning problems. We formalize this as follows:

**Definition 3.4** (Linear placement-errors). A distortion function $f$ has *linear placement-errors* (LPE) if there is a constant $L_f$ so that for any action-chain $a_0, \ldots, a_{k-1}$ executed from $(s_0, W)$ with $s_i = f(s_{i-1}, a_{i-1}, W)$, we have:
$\|f(s_0, \sum_{i=0}^{k-1} a_i, W) - s_k\| \leq L_f \cdot \sum_{i=0}^{k-1} \|a_i\|$.

Intuitively, the LPE property means that although a system distorts movements, the mismatch introduced when regrouping actions cannot grow faster than linearly with the size of the path taken. This actually includes a wide range of functions where the distortion depends on the state only:

**Proposition 3.5.** *Let $f(s, a, W) = s + g(s, W) \cdot a$ with $g : \mathcal{S} \to \mathbb{R}^{m \times d}$ a bounded matrix-function. Then $f$ has LPE with $L_f = 2 \cdot \sup_{\mathcal{S}} \|g\|$.*

As we will see, when the distortion term also depends on the action, i.e. $g(s, a, W)$, things become more involved for small actions $a$ even if $g$ is bounded and LPE does not follow without additional assumptions (see Section 5.1.1). In Proposition B.1, we introduce an even stronger property which suffices to imply LPE for distortion functions of common active positioning problems, like linear movement dynamics. More specifically, it follows directly that a linear movement-dynamics of the form $f(s, a, W) = s + Wa$ has LPE with $L_f = 0$.

Having gathered a notion of placement errors, we now need to control the stability of the value function. Specifically, even when we can precisely reach $s_j$ from $s_i$, the value function $V^\pi$ can change drastically in the vicinity of $s_j$, making it hard to guarantee that applying the accumulated action $a$ at $s_i$ is indeed beneficial. To control this, we have to impose good properties on $V^\pi$. We call a value function $V : \mathcal{S} \to \mathbb{R}$ $L_V$-*Lipschitz continuous* if for all $(s, W), (s', W) \in \mathcal{S}$ we have $|V(s, W) - V(s', W)| \leq L_V \cdot \|s - s'\|$. This is the final ingredient to prove our main statement:

**Theorem 3.6.** *Let $\pi$ be distance improving, $V^\pi$ is $L_V$-Lipschitz continuous and, let $f$ has $L_f$-placement errors. Let $(s_i, W)$ and $(s_j, W)$ on a trajectory of $\pi$ and let $a = \sum_{k=i}^{j-1} a_k$ be the sum of the chain of actions $\pi$ undertook to get from $s_i$ to $s_j$. Then $a$ is a $\pi$-shortcut for $s_i$ if*

$$\gamma \cdot V^\pi(s_j, W) - V^\pi(s_i, W) - \|s_j - s_W\|$$
$$\geq (\gamma \cdot L_V + 1) \cdot L_f \cdot \sum_{k=i}^{j-1} \|a_k\|$$

Note that if $j = i + 1$, the left hand side in Theorem 3.6 is zero. However, in that case, $a_i$ is, by definition, the only shortcut from $(s_i, W)$ to $(s_j, W)$ as its the direct connection from $s_i$ to $s_j$. Proposition 3.3 for $f(s, a, W) = s + W \cdot a$ arises as a special case of Theorem 3.6 because $L_f = 0$ implies that the right-hand side is $0$ and the left-hand side is always non-negative due to Proposition 3.1. However, Theorem 3.6 requires $V^\pi$ to be Lipschitz continuous, where no

assumptions on $\pi$ are necessary in Proposition 3.3. The next condition helps to ensure that $V^\pi$ is indeed Lipschitz continuous (see Proposition A.3), which requires a beneficial interplay with $f$:

**Definition 3.7** ($f$-contraction)**.** A policy $\pi$ is an $f$-*contraction* if for all $(s, W), (s', W)$ with respective observations with $o = O(s, W)$ and $o' = O(s', W)$, we have

$$\|f(s, \pi(o), W) - f(s', \pi(o'), W)\| \leq \|s - s'\|.$$

**Corollary 3.8.** *Let $\pi$ be distance improving $f$-contraction and let $f$ have LPE with constant $L_f$. Let $(s_i, W)$ and $(s_j, W)$ on a trajectory of $\pi$ and let $a = \sum_{k=i}^{j-1} a_k$ be the sum of the chain of actions $\pi$ undertook to get from $s_i$ to $s_j$. Then $a$ is a shortcut for $s_i$ if*

$$\gamma \cdot V^\pi(s_j, W) - V^\pi(s_i, W) - \|s_j - s_W\| \geq \frac{L_f}{1-\gamma} \cdot \sum_{k=i}^{j-1} \|a_k\|.$$

Being an $f$-contraction is a stronger requirement than mere distance improvement. We refer to Section B.2 for a discussion and examples of $f$-contractions and Lipschitz value functions in real-world policies. In practice, many active positioning policies do not satisfy the contraction property globally, yet this is not required for identifying useful shortcuts as shown in our experiments.

## 4. Trajectory Augmentation via LIFT

The idea of iterative reinforcement learning is to enrich logging policies with exploratory steps while collecting data (Zhang et al., 2023), mostly in order to improve coverage of the state-action space. Specifically, an *uncertainty model* $E_\theta(s, a)$ is trained with $E_\theta(s, \cdot)$ a probability distribution on $\mathcal{A}$ for each $s \in S$. Given a dataset $D$, $E_\theta$ is trained by minimizing $\mathbb{E}_{(o,a)\sim D}\big[-\log(E_\theta(s, a)) + \mathcal{R}(\theta)\big]$ with $\mathcal{R}(\theta)$ a regularization term. Intuitively, $E_\theta(s, a)$ can be seen as the probability that action $a$ has been seen for state $s$ in $D$. Actions with small probability $E_\theta(s, a)$ at state $s$ are considered as exploratory actions and should be selected according to some fixed probability $p$ enriching a given logging policy $\pi_\beta$ during rollout. These *exploratory actions* are rather rare and thus help keeping the system safe and naturally close to the logging policy $\pi_\beta$ that generated the data. Although this approach seems appealing, a central part has been underexplored in current literature, namely that static logging policies may not deal well with intermediate exploratory steps. In practice, arbitrary exploratory steps may lead to states from which the logging policy cannot recover well, resulting in lower overall returns. We build upon this idea, but instead of selecting actions that have not been seen in the data, we advocate to train a $Q$-function $Q_\theta$ on some initial dataset $D$ and select actions having high $Q$-values. Formally, we set $a_\theta(s, a) = \max_{a' \in \mathcal{A}} Q_\theta(s, a')$

where $Q_\theta$ can be trained with any offline RL method, like CQL or IQL. We call $a_\theta$ an *augmentor*. By that, we aim to enrich the dataset with actions that are likely to be beneficial for $\pi_\beta$ in the sense of higher returns. While this idea is quite universal and it remains unclear how actions that ease hand-offs look like in general. Moreover, in order that the augmentor provides useful steps, it has to be trained well already with limited data. The idea of LIFT is to show the augmentor data of *good behavior* by applying augmentation to the logged data that emphasizes such behavior. Clearly, when $a_\theta$ to suggest at $o = O(s, W)$ $\pi_\beta$-shortcuts (Definition 3.2), a logging policy with higher return can be obtained by combining them (see Proposition A.1 for details):

$$\pi_{\text{aug}}(o) := \begin{cases} a_\theta(o) & \text{if } a_\theta(o) \text{ is a } \pi_\beta\text{-shortcut at } (s, W) \\ \pi_\beta(o) & \text{otherwise} \end{cases}.$$

This can be seen as a specialization of the policy improvement theorem (Sutton & Barto, 2018, Section 4.2) to active positioning. For the remainder, we discuss how to train $a_\theta$ in order that it suggests $\pi_\beta$-shortcuts for active positioning problems. However, we want to emphasize that LIFT in general is not tied to this form of backbone-augmentations.

Theorem 3.6 gives a condition when and how to augment a trajectory $(o_0, a_0, r_0), \ldots, (o_n, a_n, r_n)$ with latent states $s_i = f(s_{i-1}, a_{i-1}, W)$, observations $o_i = \mathcal{O}(s_i, W)$, rewards $r_i = -\|s_{i+1} - s_W\|$, and actions $a_i = \pi_\beta(o_i)$ from a logging policy $\pi_\beta$. To convey them into a practical algorithm, let $C \in \mathbb{R}_{\geq 0}$ be a constant and let $G_i = V^{\pi_\beta}(s_i, W) = \sum_{k=i}^n \gamma^{k-i} r_k$ be the returns of $\pi_\beta$. Now, take any pair $(i, j)$ with $i < j$, let $\hat{a} = \sum_{k=i}^{j-1} a_i$ be a shortcut candidate and check if $\gamma G_j - G_i + r_{j-1} \geq C \cdot \sum_{k=i}^{j-1} \|a_k\|$ with some constant $C$ holds true. Clearly, without prior information on $f$ and $\pi_\beta$, the exact value of $C$ remains unclear, and thus it has to be considered a regularization hyperparameter of our method. If $C = 0$, all pairs are considered shortcuts, if $C$ is large, only very few pairs where high reward is gained in a few short steps are considered shortcuts. If the inequality is valid for $(i, j)$, we can assume that $\hat{a}$ is a shortcut and ideally, we would add the tuple $(o_i, \hat{a}, -\|s'_j - s_W\|, o'_j)$ with $s'_j = f(s_i, \hat{a}, W)$ and $o'_j = O(s'_j, W)$ to the dataset. However, due to the movement uncertainty, there is a gap between the position $s'_j$ the shortcut leads to and the observed state $s_j$. Particularly, the image observation $O(s'_j, W)$ and the reward $-\|s'_j - s_W\|$ differ from the actually observed ones, namely $o_j$ and $r_{j-1}$. We argue, however, that in many practical applications, this gap is small, for instance if $L_f = 0$ as in linear movement dynamics $f(s, a, W) = s + W \cdot a$ (see Proposition 3.3). Thus, we add $(o_i, a, r_{j-1}, o_j)$ to the training dataset. Algorithm 1 summarizes our shortcut sampling procedure, and we want to emphasize that it can be added to any offline RL method that samples from an offline dataset, like to minimize the Bellman error or related temporal difference errors

as in CQL. Note that for a given input tuple, the runtime of Algorithm 1 is linear in the trajectory length. Observe that the synthetic shortcuts are only used to obtain the augmentor $a_\theta$, which in turn is only used to fine-tune the logging policy, and the collected dataset consists of real data only. The precise procedure is described in Algorithm 2. For that, they must have good hand-over properties and thus we augment the dataset $D$ with shortcuts computed via Algorithm 1 when training $Q_\theta$.

---

**Algorithm 1** Shortcut sampling

---

**Require:** $C \geq 0, i \in [n], \{(o_0, a_0, r_0), \ldots, (o_n, a_n, r_n)\}$
**Ensure:** Tuple $(o_i, \hat{a}, r_{j-1}, o_j)$
 1: Compute returns $G_0 \ldots, G_n$ for trajectory
 2: $S = ()$
 3: **for** $j = i + 1 \cdots n$ **do**
 4:    $\hat{a}_i \leftarrow \sum_{k=i}^{j-1} a_k$
 5:    **if** $\gamma G_j - G_i + r_{j-1} \geq C \cdot \sum_{k=i}^{j-1} \|a_k\|$ and $\hat{a}_i \in \mathcal{A}$ **then**
 6:       Add $(o_i, \hat{a}_i, r_{j-1}, o_j)$ to $S$
 7:    **end if**
 8: **end for**
 9: Let $m = |S|$ and let $\hat{r} = (\hat{r}_1, \ldots, \hat{r}_m)$ be the rewards of the tuples in $S$
10: Let $\rho \sim \hat{r} - \min_i \hat{r}_i$ a mass function
11: Sample $(o_i, \hat{a}_i, r_{j-1}, o_j)$ from $S$ w.r.t. $\rho$
12: **return** $(o_i, \hat{a}_i, r_{j-1}, o_j)$

---

**Algorithm 2** LIFT

---

**Require:** $\pi_\beta, n \in \mathbb{N}, a_\theta, p \in [0, 1]$
**Ensure:** Dataset $D$ with $n$ trajectories
 1: Initialize $D \leftarrow \emptyset$
 2: **repeat**
 3:    Sample $o_0$ from environment
 4:    Set done = false, $\tau = (), i = 0$
 5:    **while** done is false **do**
 6:       $a_i = \pi_\beta(o_i)$
 7:       **if** rand() $\leq p$ **then**
 8:          $a_i = a_\theta(o_i, a_i)$
 9:       **end if**
10:       $o_{i+1}, r_i$, done = env.step($a_i$)
11:       Reset $\pi_\beta$ at $o_{i+1}$ (if $a_i$ was augmented by $a_\theta$)
12:       Add $(o_i, a_i, r_i)$ to $\tau, i = i + 1$
13:    **end while**
14:    Add trajectory $\tau$ to $D$
15:    **if** train augmentor **then**
16:       Train $a_\theta$ on $D$ with help of Algorithm 1
17:    **end if**
18: **until** $|D| = n$
19: **return** $D$

---

# 5. Experiments

Our experiments address two main questions: Can shortcut augmentations improve pure offline RL and can they be leveraged during data collection by training an augmentor in comparison to warm-start RL? We test different distortions $f$, observation types $\mathcal{O}$, and levels of logging expertness.

## 5.1. Environments

In order to analyze different movement distortions and observation types in isolation, we conducted our experiments in semi-realistic active positioning environments designed to keep real world characteristics and entail small sim-to-real gaps. Throughout, we use $-\|s - s_W\|$ as reward signal, which is easy to compute in simulations, as one typically has access to latent information $(s, W)$. When data is coming from a real system, In real systems, this signal can easily be added in hindsight to finished episode once $s_W$ is uncovered by the logging policy.

### 5.1.1. MOVEMENT DISTORTIONS

We consider different movement distortions, some of them have linear forms, like $f_{\text{blend}}$ and $f_{\text{rot}}$ both with $L_f = 0$. We also use non-linear distortions, like $f_{\text{scale}}$ and $f_{\text{sin}}$ which have LPE with $L_f > 0$ and one non-continuous distortion $f_{\text{regrot}}$ also having LPE which is not contracting. Moreover, we test a dynamics $f_{\text{sqrt}}$ that does not satisfy the LPE property. We refer to Section B for their precise mathematical definitions and proofs of their properties. Figure 4 illustrates an overview of the different distortions in two dimensions.

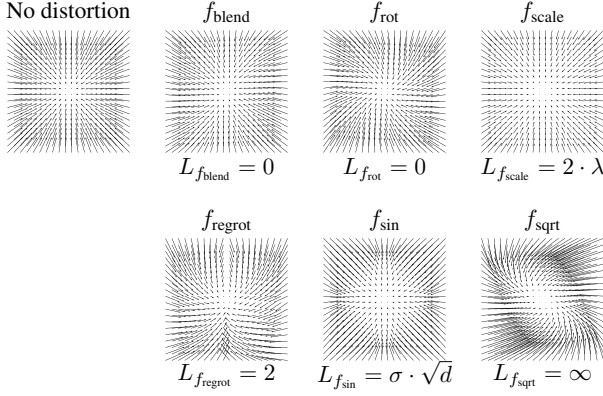

*Figure 4.* Movement distortions used when applying actions $\text{clip}_\lambda(s_W - s)$.

### 5.1.2. OBSERVATIONS

A canonical type of observation is when the position can be observed directly, i.e., $\mathcal{O}_{\text{PO}}(s, W) = s$. Here, we need to fix the optimum $s_W = s^*$, because it is impossible to infer $s_W$ without observing $W$ (see also Section C). Roughly

speaking, these are scenarios where it is known where the optimum is, but not how to get there. We will evaluate these scenarios in $d = 2$ and $d = 5$ dimensions. Our motivation stems from scenarios where observations are drawn from optical sensors and hence we test our method on different image generators (Figure 5). The first comes from active alignments problems from camera assembly, where a lens objective has to be positioned relative to a sensor to obtain optimal optical performance (Liu et al., 2024). Here, $s$ relates to the position of the lens objective and $W$ to variances in the lenses of the objective and distortions in the movement dynamics. At each position $s$, light is sent through the lens system creating an image $\mathcal{O}_{\mathrm{LP}}(s, W)$ on a sensor. The task is to position the objective with variances $W$ precisely to an individual optimum $s_W$ (Figure 2) As some information about $W$ is contained in the image implicitly, it is possible to design algorithms that leverage the image information to move towards $s_W$. We use the realistic generator from (Burkhardt et al., 2025) where light is sent in the form of a *Siemens star* producing images whose contrast and sharpness are sensitive to small misalignments.

We also run experiments in the *Fetch Reach* environments (Plappert et al., 2018), where a robotic arm has to reach a desired position $s_W$. Here, we use the vanilla environment $\mathcal{O}_{\mathrm{Fetch}}(s, W) = s - s_W$ where the distance to the target is observed. In Section D we study the effect of shortcut augmentation for harder variants using image observations $\mathcal{O}_{\mathrm{FetchImg}}$ and reaching multiple goals subsequently from offline data alone.

Our last image generator is the *light tunnel* from (Gamella et al., 2025), where light is sent through two polarizers whose angles dictate how it passes through to an optical sensor. Here, each position $s$ of the polarizers filters out certain wavelengths of the light creating a image $\mathcal{I}(s)$ at the sensor. Here, $\mathcal{I}(s)$ does not depend on the context $W$ but only on the relative difference of the angles of the polarizers, i.e. many states lead to the same image. To add context, we sample in each episode $s_W$ uniformly from the box $[0, 2\pi]^2$ and set $\mathcal{O}_{\mathrm{LT}}(s, W) = \mathcal{I}(s) - \mathcal{I}(s_W)$. In our experiments, we use the decoder of the autoencoder trained on images from the real system provided in the data repository of (Gamella et al., 2025).

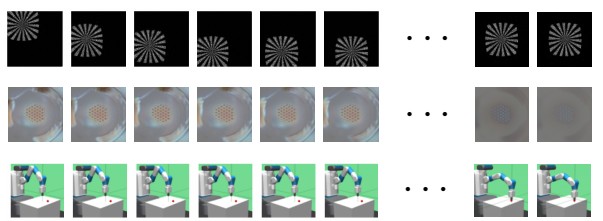

*Figure 5.* Exemplary trajectories of $\pi_{\mathrm{cw},l}$ executed in $\mathcal{O}_{\mathrm{LP}}$, $\mathcal{O}_{\mathrm{LT}}$, and $\mathcal{O}_{\mathrm{FetchImg}}$ (top to bottom).

### 5.1.3. LOGGING POLICIES

In most offline RL benchmarks, logging policies are obtained by training online RL algorithms partially or fully to obtain policies of different expertise (Fu et al., 2021). However, in many real-world continuous-control settings, logging policies are hand-crafted, highly structured, and systematically suboptimal routines. This is particularly common in active positioning tasks, where expert routines rely on relatively simple mechanisms yet can be applied across a wide range of systems with only minimal adjustments. A representative example are optical alignment procedures, in which system performance is improved iteratively by sequentially adjusting individual degrees of freedom and evaluating a measured signal such as coupling efficiency or spot quality (Parks, 2006; An et al., 2021; Langehanenberg et al., 2015). Similar principles also apply to other positioning and manipulation tasks that use coordinate-based or heuristic search strategies. These methods usually start with rough movements and reduce the step size over time until the target is reached. (Liu et al., 2024, Section 3.1). To study offline RL under such structured but imperfect data in a controlled and reproducible manner, we require a logging policy that reliably reaches the target while producing trajectories that are suboptimal in both direction and number of steps, and whose expertness can be varied systematically. We distill these principles into a synthetic logging policy referred to as the *coordinate walk* $\pi_{\mathrm{cw},l}$.

Across all scenarios we study, successful control requires that relevant displacement information — essentially $s - s_W$ — is inferable from the observation $\mathcal{O}(s, W)$, since otherwise the task is not solvable. Even when $s - s_W$ is inferable, however, the task may still be unsolvable without any information about the movement distortion $f$. This requirement is discussed more formally in Appendix C and instantiated concretely in Section 5.1.2 for the various observation settings we study. To generate reliable trajectories across these different observation settings and distortion regimes, we gave the logging policy direct access to $s - s_W$. Importantly, this does not make the task trivial, we simply assume the logging policy already has a reliable way to infer the relevant information from $\mathcal{O}(s, W)$. Inspired by real-world logging policies as described above, we constructed a structured logging policy that optimizes coordinate by coordinate. That is, actions are chosen along coordinate axes until the corresponding coordinate of $s$ matches that of $s_W$. Once all dimensions have been traversed, the step size $l$ is reduced and the procedure is repeated, resulting in a reliable but As a result, the logging policy can reach the target for the movement distortions we consider, but it does so highly inefficiently, including overshoots, detours, and movement in the wrong direction. In Section E.4, we show that our method is not dependent on structured logging policies.

By varying the initial step size, the expertness of the logging policy can be adjusted (see Figure 10). Figure 11 shows trajectories of the coordinate walk executed under different movement distortions. To model realistic hand-overs between logging policies and augmentors, we assume the internal state of the policy, i.e. the current step size $l$ and dimensions already optimized, is reset to the initial values once the policy is reset. To avoid making our mathematical framework introduced in Section 3 too specific for these types of resets, we assume stateless policies there. For most states, $V^{\pi_{l_2}}(s, W) \geq V^{\pi_{l_1}}(s, W)$ for two step sizes $l_1 < l_2$ holds true and thus Theorem 3.6 holds in this setting. In Section B.2, a detailed discussion on the contraction-property and LPE of $\pi_{\mathrm{cw},l}$ is given.

## 5.2. Results

Section 4 gives rise to two algorithms. First, a purely offline one that takes a static dataset collected from some logging policy and trains an offline RL algorithm with shortcut augmentations. In our experiments, we use CQL and denote this algorithm as CQL-SC. Second, an iterative offline RL algorithm that collects data with an augmented logging policy where CQL is trained on the collected data, called LIFT. If the subsequently trained CQL also uses shortcuts, we denote this algorithm as LIFT-SC. By default, we use Algorithm 2 with $p = 0.6$, limit augmentations per trajectory to 20. In Section E.5, we study in detail the sensitivity of our method to the choice of the hyperparameter $C$. Larger values of $C$ are more restrictive in terms of which augmentations are sampled. Although better policies can be obtained by tuning $C$, particularly when $L_f$ is comparatively large like in $f_{\mathrm{regrot}}$, we set $C = 0$ in all experiments to ensure a fair comparison and to avoid introducing additional inductive biases into our method. A detailed hyperparameter analysis is given in Section E.1.

First, we analyze the effect of different augmentations while collecting data and the effect of using shortcuts in the CQL training afterward. Beside naive augmentations as adding gaussian noise $\pi_\beta(o) + \epsilon$ or randomly scaling actions $\pi_\beta(o) \cdot \epsilon$ with $\epsilon = 2 \cdot \exp(\eta), \eta \sim \mathcal{N}(0, \sigma)$, we also use uniformly sampled actions from $\mathcal{A}$ and IORL-like augmentations based on an uncertainty model as in (Zhang et al., 2023). We run these experiments in $(\mathcal{O}_{\mathrm{PO}}, f_{\mathrm{blend}})$ with step size 0.025 in $d = 5$ dimensions, collected 3 independent datasets consisting of 100 trajectories each and trained 3 independent CQL policies on each of them. The LIFT augmentor is trained once after 50 trajectories. The averaged convergences to $s_W$ of the CQL policies, each evaluated on 20 randomly drawn contexts are shown in Figure 6a. Once can see that independently whether shortcuts are used in the training afterward, the best CQL policies is obtained when trained on the data collected with LIFT. Moreover, we see that when training takes place with shortcuts, every policy

can be improved. This finding is underpinned when computing the dataset characteristics introduced in (Schweighofer et al., 2022) shown in Figure 6b. LIFT creates trajectories having the highest average returns reproducing findings in (Schweighofer et al., 2022) that this correlates with CQL performance. On the other hand, LIFT does not explore as well as other methods, showing a clear differentiation to IORL that has been explicitly designed to explore well. However, high exploration comes at the price of an impeded hand-off back to the logging policy, leading to low trajectory qualities for IORL and random actions.

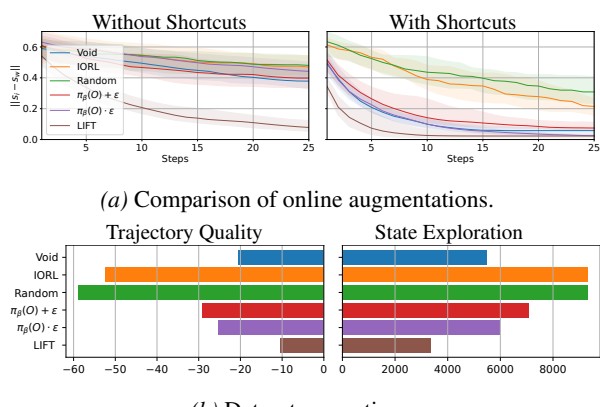

*(a)* Comparison of online augmentations.

*(b)* Dataset properties.

*Figure 6.* Experiments in $(\mathcal{O}_{\mathrm{PO}}, f_{\mathrm{blend}})$ with $l = 0.025$ and $d = 5$.

In our second type of experiments, we evaluate how our methods compare under different movement distortions and observation types. In $\mathcal{O}_{\mathrm{PO}}$, algorithms collect a total of $n = 100$ and $n = 500$ trajectories for $d = 2$ and $d = 5$ respectively, where the LIFT augmentor is trained once after 50 and 100 collected trajectories respectively. In $\mathcal{O}_{\mathrm{LP}}$, we collect 500 trajectories and LIFT is trained once after 100 episodes. In $\mathcal{O}_{\mathrm{LT}}$, we collect only 100 trajectories and LIFT is trained once after 50 collected trajectories. Here, we additionally compare to SAC (Haarnoja et al., 2018) trained with a mixture of offline and online data as done in warm-start RL that is restricted to the same number of trajectories as in our offline datasets. Specifically, in a scenario with $n$ episodes, the replay buffer of SAC is initialized with the same number of trajectories collected by the logging policy the LIFT augmentor obtains in training, e.g. $m = 50$ for $\mathcal{O}_{\mathrm{LT}}$. Moreover, we also compare to diffusion-based techniques, like GTA (Lee et al., 2024) that generate synthetic transitions and Diffusion-QL (DQL) (Wang et al., 2023) that learns a diffusion-based policy. Figure 7 presents selected comparisons across the multiple scenarios and all comparisons can be found in Section E. In all tested environments, we see that CQL policies trained offline on data from LIFT have better performance than these trained on unaugmented data from the logging policy. This effect fades a bit when adding shortcuts to the subsequent offline training: In most scenarios, the performance of LIFT-SC is better or equal

than CQL-SC. This is, for instance, not the case when using image data from $\mathcal{O}_{\mathrm{LP}}$, where CQL training on data obtained from LIFT-SC showed high variance. Studying the effect of shortcuts in isolation, CQL-SC consistently outperforms CQL and LIFT-SC consistently outperforms LIFT, making LIFT-SC the best of our methods. Comparing LIFT-SC to SAC with offline data, we see a clear picture: SAC stays ahead in all low-dimensional cases for $\mathcal{O}_{\mathrm{PO}}$, and LIFT-SC outperforms SAC almost consistently over all movement dynamics and expert-levels of the logging policy in $\mathcal{O}_{\mathrm{PO}}$ for $d = 5$ (see Appendix E.3), as well as in image-based scenarios. Interestingly, for $f_{\mathrm{regrot}}$ where the contraction property is violated, augmentations with shortcut fail, whereas in $f_{\mathrm{sqrt}}$, where LPE does not hold, augmentations still help but the advantage over SAC is negligible.

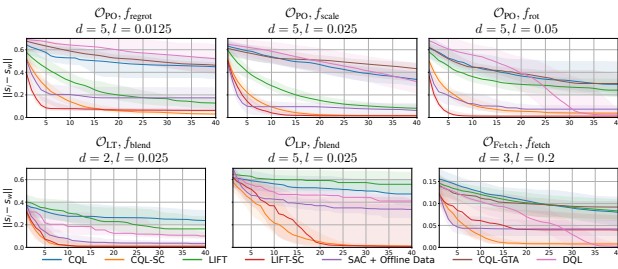

*Figure 7.* Comparisons of our methods for selected scenarios.

Finally, we analyse the effect of absence of structure in the logging policy on the performance of the shortcut augmentation by injecting noise into the $\pi_{\mathrm{cw},l}$. The results are in presented in Section E.4 and in the tested scenarios, we found that shortcut augmentation consistently yields better policies, suggesting that benefits of shortcuts are not limited to structured logging policies.

## 6. Discussion

We demonstrate that shortcut augmentations can consistently improve the effectiveness of offline RL in active positioning problems in both, theoretical and experimental validations. In particular, we find that augmentations provide the largest gains in complex scenarios with higher action dimensionality or partial observability, where plain offline RL often fails. This suggests that exploiting task structure to expand data coverage is a promising alternative to relying solely on behavior regularization. Compared to warm-start RL, LIFT offers a more data-efficient way to leverage suboptimal expert routines: by selectively taking shortcuts suggested by an off-policy learner, we improve dataset quality without requiring extensive online fine-tuning. Nevertheless, our approach has limitations. Shortcut validity depends on assumptions about the distortion function and value function regularity, which may not hold in all real-world positioning systems. Moreover, our experiments are limited to semi-

realistic simulators; future work should validate these methods on physical platforms, especially in robotic alignment tasks. Another open question is how to combine shortcut augmentation with model-based methods or world models to further improve sample efficiency. We believe that the principles underlying LIFT are broadly applicable beyond the scenarios studied in in this work where expert routines exist but are suboptimal. We hope this work encourages a more systematic treatment of data augmentation strategies for offline RL in structured industrial tasks.

## Acknowledgments

This research was funded by the German Federal Ministry of Research, Technology and Space (BMFTR) under grant number 13FH605KX2. TW is funded by the *Hightech Agenda Bavaria*. We thank our colleagues Michael Layh and Martin Wenzel for helpful discussions and feedback on the manuscript.

## Impact Statement

This paper presents work whose goal is to advance the field of Machine Learning. There are many potential societal consequences of our work, none which we feel must be specifically highlighted here.

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

# A. Proofs for Section 3

**Proposition A.1.** *Let $\pi_\beta$ and $a_\theta$ be two policies and $o = O(s, W)$, then $J(\pi_{aug}) \geq J(\pi_\beta)$ with $\pi_{aug}$ defined as follows:*

$$\pi_{\text{aug}}(o) := \begin{cases} a_\theta(o) & \text{if } a_\theta(o) \text{ is a } \pi_\beta\text{-shortcut at } (s, W) \\ \pi_\beta(o) & \text{otherwise} \end{cases} .$$

*Proof of Proposition A.1.* We denote $\pi_\beta$ simply by $\pi$ in the following. It suffices to show that the statement holds if augmentation only is applied at one single state $(\tilde{s}, W)$ as we than can apply the statement repeatedly. That is, there exists an action $a$ that satisfies:

$$\gamma \cdot V^\pi(f(\tilde{s}, a, W), W) - \|f(\tilde{s}, a, W) - s_W\| \geq V^\pi(\tilde{s}, W)$$

Let $\pi_a$ be the policy that uses $a$ at $\tilde{s}$ and on all other states coincides with $\pi$. First, we show that $J(\pi_a) \geq J(\pi)$. It suffices to show that $V^{\pi_a}(s) \geq V^\pi(s)$ for all $s \in S$. Let $(s, W)$ be an initial state. If the trajectory of $\pi$ does not traverse $\tilde{s}$, then $V^{\pi_a}(s) = V^\pi(s)$. Assume differently that the trajectory visits $\tilde{s}$ at the $t$-th step. Then, the trajectory starting at $s$ follows $\pi$ till $\tilde{s}$, then chooses the shortcut $a$, and then follows $\pi$ from $s' = f(\tilde{s}, a, W)$. The value for this trajectory is:

$$V^{\pi_a}(s, W) = V^\pi(s, W) - \gamma^t \cdot V^\pi(\tilde{s}, W) - \gamma^t \|s' - s_W\| + \gamma^{t+1} V^\pi(s', W).$$

From the assumption of $(\tilde{s}, a)$, we have

$$\gamma^t \cdot (-V^\pi(\tilde{s}, W) - \|s' - s_W\| + \gamma \cdot V^\pi(s', W)) \geq 0$$

and hence $V^{\pi_a}(s, W) \geq V^\pi(s, W)$. □

**Lemma A.2.** *Let $\pi$ be distance-improving, then $(1 - \gamma)V^\pi(s, W) \geq -\|s - s_W\|$ for all $(s, W)$.*

*Proof.* Let $(s_0, W), (s_1, W), \ldots, (s_k, W)$ be a trajectory of $\pi$ starting at $s = s_0$, then

$$V^\pi(s, W) = -\sum_{i=1}^{k} \gamma^{i-1} \|s_i - s_W\| \geq -\|s - s_W\| \sum_{i=0}^{k-1} \gamma^i = -\|s - s_W\| \cdot \frac{1 - \gamma^k}{1 - \gamma}$$

where we have used that $\pi$ is distance improving in every step. Finally, $(1 - \gamma)V^\pi(s, W) \geq -\|s - s_W\|(1 - \gamma^k) \geq -\|s - s_W\|$. □

*Proof of Proposition 3.1.* Assume that $\tau = (s_0, \ldots, s_k)$ is the sub-trajectory of $\pi$ starting at $s = s_0$ and ending at $s' = s_k$. We prove the statement via induction on $k$. Note that since $s' \neq s$, we have $k \geq 1$. Let $k = 1$, then

$$V^\pi(s, W) = -\|s_1 - s_W\| + \gamma \cdot V^\pi(s', W)$$

and the claim holds. Now, assume the statement holds from $s_1$ to $s_k = s'$, then

$$\gamma V^\pi(s', W) - V^\pi(s_1, W) \geq \|s' - s_W\|$$

by the induction hypothesis. Furthermore, we have

$$\begin{aligned} \gamma V^\pi(s', W) - V^\pi(s, W) &= \gamma V^\pi(s', W) - V^\pi(s_1, W) + V^\pi(s_1, W) - V^\pi(s, W) \\ &\geq \|s' - s_W\| + V^\pi(s_1, W) - V^\pi(s, W) \\ &= \|s' - s_W\| + V^\pi(s_1, W) - (-\|s_1 - s_W\| + \gamma V^\pi(s_1, W)) \\ &= \|s' - s_W\| + (1 - \gamma)V^\pi(s_1, W) + \|s_1 - s_W\| \end{aligned}$$

Using Lemma A.2, we have $(1 - \gamma)V^\pi(s_1, W) + \|s_1 - s_W\| \geq 0$ and the claim follows. □

*Proof of Proposition 3.3.* Since Proposition 3.1 gives that $\gamma V^\pi(s_j, W) - V^\pi(s_i, W) \geq \|s_j - s_W\|$, it is left to prove that $f(s_i, a, W) = s_j$. We have

$$f(s_i, a, W) = s_i + W \cdot \sum_{k=i}^{j-1} a_i = s_i + W \cdot a_i + W \cdot a_{i+1} + \ldots + W \cdot a_{j-1}.$$

Let $s_{i+1}, \ldots, s_{j-2}$ be the intermediate states, i.e. $s_k = f(s_{k-1}, a_{k-1}, W)$, then replacing $s_k = s_k - 1 + W \cdot a_{k-1}$ in the equation above from $k = i$ to $k = j - 1$ gives the claim. □

*Proof of Proposition 3.5.* Let $a_0, \ldots, a_{k-1}$ a chain of actions and set $A = \sum_{i=0}^{k-1} = a_i$, $(s_0, W)$ an initial state and set $s_i = f(s_{i-1}, a_{i-1}, W)$. Recursively unraveling the definition of $f$ yields

$$s_k = s_0 + \sum_{i=0} g(s_i, W) \cdot a_i$$

and consequently

$$f(s_0, A, W) - s_k = g(s_0, W) \sum_{i=0}^{k-1} a_i - \sum_{i=0}^{k-1} g(s_i, W))a_i$$
$$= \sum_{i=0}^{k-1} \big(g(s_0, W) - g(s_i, W)\big)a_i.$$

Taking norms and using the induced matrix norm on $\mathbb{R}^{m \times d}$ gives

$$\big\|f(s_0, A, W) - s_k\big\| \leq \sum_{i=0}^{k-1} \big\|g(s_0, W) - g(s_i, W)\big\| \cdot \|a_i\|.$$

By the assumption on $g$, we have

$$\|g(s_0, W) - g(s_i, W)\| \leq \|g(s_0, W)\| + \|g(s_i, W)\| \leq 2 \cdot \sup_{\mathcal{S} \times \mathcal{W}} \|g\|$$

independently of the actions for all $i$ and the claim follows. □

*Proof of Theorem 3.6.* For brevity, we omit $W$ in the notation of the value function. We have to show that $\gamma V^\pi(f(s_i, a, W)) - V^\pi(s_i) \geq \|f(s_i, a, W) - s_W\|$. Because $f$ has linear-placement errors, it follows directly from Definition 3.4 that $\|f(s_i, a, W) - s_j\| \leq L_f \cdot \sum_{k=i}^{j-1} \|a_k\|$ and thus

$$\|f(s_i, a, W) - s_W\| = \|f(s_i, a, W) - s_j + s_j - s_W\| \leq L_f \cdot \sum_{k=i}^{j-1} \|a_k\| + \|s_j - s_W\|.$$

On the other hand, using the Lipschitz continuity of $V^\pi$, we get

$$\gamma V^\pi(f(s_i, a, W)) - V^\pi(s_i) \geq \gamma \cdot (V^\pi(s_j) - L_V \cdot \|f(s_i, a, W) - s_j\|) - V^\pi(s_i)$$
$$\geq \gamma \cdot V^\pi(s_j) - V^\pi(s_i) - \gamma \cdot L_V \cdot L_f \cdot \sum_{k=i}^{j-1} \|a_k\|$$

Now, as the inequality from the theorem statement holds, we have

$$\gamma \cdot V^\pi(s_j) - V^\pi(s_i) \geq (\gamma \cdot L_V + 1) \cdot L_f \cdot \sum_{k=i}^{j-1} \|a_k\| + \|s_j - s_W\|$$

and plugging this into the upper equation gives the claim. □

**Proposition A.3.** *Let $\pi$ be an $f$-contraction. Then $V^\pi$ is $\frac{1}{1-\gamma}$-Lipschitz continuous in the states.*

*Proof.* Define $L = \frac{1}{1-\gamma}$ and let $(s, W)$ and $(s', W)$ be two states. We prove via induction over the combined number of steps $k$ needed to reach the optimality region around $s_W$ starting at $s$ and $s'$ that

$$|V^\pi(s, W) - V^\pi(s', W)| \leq L \cdot \|s - s'\|.$$

If $k = 0$, then $s$ and $s'$ are both within the optimality region, i.e. $\|s - s_W\| \leq \theta$ and $\|s' - s_W\| \leq \theta$, then $V^\pi(s, W) = V^\pi(s', W) = 0$ and the claim holds. Now, let $o = O(s, W)$ and $o' = O(s', W)$ be the observations at $s$ and $s'$ and $s_1 = f(s, \pi(o), W)$ and $s_1' = f(s', \pi(o'), W)$ be the next states after one step of $\pi$. Particularly, the induction hypothesis holds for $s_1$ and $s_1'$, i.e. $|V^\pi(s_1, W) - V^\pi(s_1', W)| \leq L \cdot \|s_1 - s_1'\|$. Since $V^\pi(s) = -\|s_1 - s_W\| + \gamma V^\pi(s, W)$ and $V^\pi(s') = -\|s_1' - s_W\| + \gamma V^\pi(s', W)$, we have

$$
\begin{aligned}
|V^\pi(s) - V^\pi(s')| &= |\gamma \cdot V^\pi(s_1, W) - \gamma \cdot V^\pi(s_1', W) - \|s_1 - s_W\| + \|s_1' - s_W\| | \\
&\leq \gamma \cdot |V^\pi(s_1, W) - V^\pi(s_1', W)| + |\|s_1 - s_W\| - \|s_1' - s_W\| | \\
&\leq \gamma \cdot L \cdot \|s_1 - s_1'\| + \|s_1 - s_1'\| \\
&\leq (\gamma \cdot L + 1) \cdot \|s_1 - s_1'\| \\
&= L \cdot \|s_1 - s_1'\|
\end{aligned}
$$

where the last equation is due to $L = \frac{1}{1-\gamma}$. Finally, because $\pi$ is an $f$-contraction, we have $\|s_1 - s_1'\| = \|f(s, \pi(o), W) - f(s', \pi(o'), W)\| \leq \|s - s'\|$ and the claim follows. $\quad\square$

*Proof of Corollary 3.8.* Because $\pi$ is an $f$-contraction, $V^\pi$ is $\frac{1}{1-\gamma}$-Lipschitz continuous by Proposition A.3. Plugging $L_V = \frac{1}{1-\gamma}$ into Theorem 3.6 gives the claim. $\quad\square$

# B. Movement distortion functions

In this section, we formally define the different movement distortions $f$ we consider in our experiments. The first set of distortions are linear distortions of the form $f(s, a, W) = s + W \cdot a$ with $W \in \mathbb{R}^{d \times d}$ a distortion matrix, more specific, we use

$$f_{\text{blend}}(s, a, W) = s + (I_{d \times d} + W) \cdot a, \quad W \sim \mathcal{N}_{d \times d}(0, \sigma)$$

For $W \in \mathbb{R}$ a scalar, let $R_W = \begin{pmatrix} \cos(W) & -\sin(W) \\ \sin(W) & \cos(W) \end{pmatrix}$ be a two-dimensional rotation matrix. We rise this to a high-dimensional rotation matrix where adjacent dimensions are rotated, i.e.,

$$\text{Rot}_W = \text{diag}(R_W, \ldots, R_W) \in \mathbb{R}^{d \times d}$$

where $\text{diag}(A_1, \ldots, A_k)$ is the block-diagonal matrix with blocks $A_1, \ldots, A_k$ on the diagonal.

$$f_{\text{rot}}(s, a, W) = s + \text{Rot}_W \cdot a, \quad W \sim \mathcal{N}(0, \sigma)$$

The next distortion function is a scaling-based one which does not depend on a latent context $W$:

$$f_{\text{scale}}(s, a, W) = s + \text{clip}_{C, \lambda}(\|s - s_W\|) \cdot a$$

with some constant $0 < C < \lambda$ to ensure that the steps are not to small so that the optimum can be reached in finitely many steps.

The next set of distortions is again a rotation-based one, but one where the rotation matrix depends on the region. For that, we assume the position space $\mathcal{P}$ is decomposed into $c$-many non-overlapping subsets $\mathcal{P}_1, \ldots, \mathcal{P}_c$ such that $\cup_{i=1}^c \mathcal{P}_i = \mathcal{P}$. Then

$$f_{\text{regrot}}(s, a, W) = s + \sum_{i=1}^{c} \mathbf{1}_{s \in \mathcal{P}_i} \cdot \text{Rot}_{W_i} \cdot a, \quad W \in \mathcal{N}_c(\mu, \sigma), \mu \in \mathbb{R}^c$$

As $\mathcal{P}_i \cap \mathcal{P}_j = \emptyset$ for $i \neq j$, only one rotation matrix is active at a time, depending on the state.

In our experiments, we used $c = 4$ and divided $\mathcal{P}$ into four sets depending on in which quadrant of $\mathbb{R}^2$ the first two dimensions reside. Moreover, we set $\mu = (-0.3, 0.6, -0.3, 0.6)$.

The next distortion is one where a non-linear offset is added which depends on both, the state and the action:

$$f_{\sin}(s, a, W) = s + a + W \cdot \sin(s) \circ \cos(s) \cdot \|a\|, \quad W \sim \mathcal{U}(0, \sigma)$$

where $\sin$ and $\cos$ are applied component-wise and $\circ$ denote the element-wise multiplication. Finally, we consider a distortion function that does not have linear placement errors:

$$f_{\text{sqrt}}(s, a, W) = s + (I_{d \times d} + W) \cdot \sqrt{\|a\|} \cdot a, \quad W \sim \mathcal{N}_{d \times d}(0, \sigma).$$

### B.1. Linear placement-errors

We begin by proving a stronger conditions, which is easier to check and implies LPE:

**Proposition B.1.** *Let $f$ be a distortion function and assume there exists a constant $L_f$ such that for all states $(s, W)$ and actions $a, a' \in \mathcal{A}$*

$$\|f(s, a + a', W) - f(f(s, a, W), a', W)\| \leq L_f \cdot \|a\|$$

*Then $f$ has LPE with constant $L_f$.*

*Proof.* For $i \in \{0, \ldots, k\}$, define the tail sums $\tilde{a}_i := \sum_{j=i}^{k-1} a_j$ and the states $\tilde{s}_i := f(s_i, \tilde{a}_i, W)$. By definition $\tilde{s}_0 = f(s_0, a_0 + \ldots + a_{k-1}, W)$ and, since $\tilde{a}_k = 0$ and $f(s, 0, W) = s$, we also have $\tilde{s}_k = s_k$. Thus, we have to prove that $\|\tilde{s}_0 - \tilde{s}_k\| \leq L_f \sum_{i=0}^{k-1} \|a_i\|$. Now, for any $i \in \{0, \ldots, k-1\}$ we have

$$\|\tilde{s}_i - \tilde{s}_{i+1}\| = \|f(s_i, a_i + \tilde{a}_{i+1}, W) - f(s_{i+1}, \tilde{a}_{i+1}, W)\| \leq L_f \|a_i\|.$$

because of the assumptions on $f$ from the statement of the proposition. Summing these inequalities and applying the triangle inequality yields

$$\|\tilde{s}_0 - \tilde{s}_k\| \leq \sum_{i=0}^{k-1} \|\tilde{s}_i - \tilde{s}_{i+1}\| \leq L_f \sum_{i=0}^{k-1} \|a_i\|.$$

$\square$

LPE and the proposition of Proposition B.1 are not equivalent: Consider $f(s, a) = s + \text{sign}(s) \cdot a$. Then its easy to show that $f$ has linear-placement errors with $L_f = 2$, but it does not have the property from Proposition B.1.

**Proposition B.2.** *The distortion $f_{\text{blend}}$ has LPE with $L_{f_{blend}} = 0$.*

*Proof.* Straight-forward application of Proposition B.1. $\square$

**Proposition B.3.** *The distortion $f_{\text{rot}}$ has LPE with $L_{f_{rot}} = 0$.*

*Proof.* Straight-forward application of Proposition B.1. $\square$

**Proposition B.4.** *The distortion $f_{\text{scale}}$ has LPE with $L_{f_{scale}} = 2 \cdot \lambda$.*

*Proof.* We write $f_{\text{scale}}(s, a, W) = s + g(s, W) \cdot a$ with $g(s, W) = \text{clip}_{C,\lambda}(\|s - s_W\|) \cdot I_d$ with $I_d$ the identity function of $\mathbb{R}^{d \times d}$. Clearly $g$ is bounded and we have $\sup_{\mathcal{S} \times \mathcal{W}} \|g\| = \lambda$ and the claim follows by an application of Proposition 3.5. $\square$

**Proposition B.5.** *The distortion $f_{\text{regrot}}$ has LPE with $L_{f_{regrot}} = 2$.*

*Proof.* We write $f_{\text{regrot}}(s, a, W) = s + g(s, W) \cdot a$ with $g(s, W) = \text{Rot}_{W_i}$ whenever $s \in \mathcal{P}_i$, where $\mathcal{P}_1, \ldots, \mathcal{P}_c$ are the partitions of $\mathcal{S}$ from Section 5.1.1. For every state $(s, W)$, $g(s, W)$ is a rotation matrix and thus $\|g(s, W)\| = 1$ and $g$ statisfies the the claim follows from Proposition 3.5. □

**Proposition B.6.** *The distortion* $f_{\sin}$ *has LPE with* $L_{f_{sin}} = \sqrt{d}\sigma$.

*Proof.* Let $f_{\sin}(s, a, W) = s + a + g(s) \cdot \|a\|$ with $g(s, W) := W \cdot \sin(s) \odot \cos(s)$. Although we cannot apply Proposition 3.5 as $f_{\sin}$ has not the desired form, we can follow a similar strategy. First, we observe that $g$ is bounded:

$$\|g(s, W)\| = |W| \cdot \sqrt{\sum_{i=1}^{d} \sin(s_i)^2 \cdot \cos(s_i)^2} \leq \sigma\sqrt{d}$$

because $W \sim \mathcal{U}(0, \sigma)$. Let $a_0, \ldots, a_{k-1}$ be a chain of actions and set $A = \sum_{i=1}^{k-1} a_i$ and $s_i = f(s_{i-1}, a_{i-1}, W)$, then

$$f_{\sin}(s_0, A, W) - s_k = A + g(s_0, W)\|A\| - \sum_{i=0}^{k-1}\Big(a_i + g(s_i, W)\|a_i\|\Big) = g(s_0, W)\|A\| - \sum_{i=0}^{k-1} g(s_i, W)\|a_i\|$$

and thus:

$$\|f_{\sin}(s_0, A, W) - s_k\| \leq \|g(s_0, W)\|A\| + \sum_{i=0}^{k-1} \|g(s_i, W)\|a_i\| \leq \sigma\sqrt{d}\sum_{i=0}^{k-1}\|a_i\|$$

because $\|A\| \leq \sum_{i=0}^{k-1}\|a_i\|$ by the triangle inequality. □

Next, we show that $f_{\text{sqrt}}$ is not LPE:

**Proposition B.7.** *The distortion* $f_{\text{sqrt}}$ *does not have LPE.*

*Proof.* Let $v \in \mathbb{R}^d$ be a unit vector and let $a_0 = a_1 = c \cdot v$ with $c \leq \lambda$. Let $(0, 0) \in \mathbb{R}^d \times \mathbb{R}^{d \times d}$ be an initial state, then $s_1 = f_{\text{sqrt}}(0, a_0, 0) = \sqrt{c} \cdot c \cdot v$ and $s_2 = f_{\text{sqrt}}(s_1, a_1, 0) = 2\sqrt{c} \cdot c \cdot v$. Moreover, we have $f(s_0, a_0 + a_1, 0) = f(0, 2 \cdot c \cdot v, 0) = 2\sqrt{2c} \cdot c \cdot v$ and hence

$$\|f(s_0, a_0 + a_1, 0) - s_2\| = (2\sqrt{2} - 2) \cdot \sqrt{c} \cdot c.$$

which cannot be bounded by $L_f \cdot (\|a_0\| + \|a_1\|) = 2 \cdot L_f \cdot c$ for any constant $L_f$. □

## B.2. Contractions and Lipschitz-continuity in real-world applications

We do not expect that policies and distortions from real-world applications satisfy the rigorous mathematical assumptions stated in Section 3. Pedantically, even simple modeling choices already break global smoothness: for instance, having $\mathcal{A} = B_\lambda(0)$ with $\mathcal{A}$ a strict subset of $S$, combined with an optimality region defined by a threshold $\theta$, induces discontinuities in the value function. The same holds for the coordinate walk policy in Section 5.1.3, where a fixed step length produces value functions with sharp discontinuities, as shown in Figure 9.

Nevertheless, global mathematical rigor is not required to detect local shortcuts in real trajectories. A striking example is the coordinate walk under $f_{\text{regrot}}$: since different rotations apply in different regions, the policy is not an $f$-contraction globally, because nearby states $s$ and $s'$ lying in different regions $\mathcal{P}_i$ and $\mathcal{P}_j$ may be rotated in different directions (Figure 8a). Yet, for states within same region where the coordinate walk applies same actions, the contraction property is preserved (Figure 8b). This illustrates that shortcut identification relies less on global guarantees and more on local structure along trajectory segments.

Informally speaking, it suffices that the value function does not change too abruptly for small misplacements, so that local improvements can be exploited as shortcuts. In practice, this condition is often met: physical systems typically exhibit continuity over small ranges of motion, even if discontinuities or non-contractive behavior emerge globally. Hence, while our theoretical assumptions provide clean guarantees, the underlying ideas remain applicable well beyond the idealized setting as demonstrated by our experiments in Section 5.

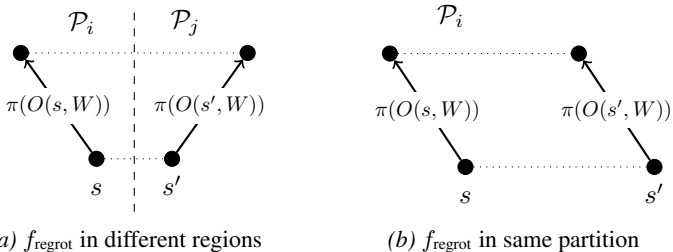

(a) $f_{\text{regrot}}$ in different regions    (b) $f_{\text{regrot}}$ in same partition

*Figure 8.* In $f_{\text{regrot}}$, starting at two close-by states $s$ and $s'$ in different regions $\mathcal{P}_1$ and $\mathcal{P}_2$ can increase the distance between subsequent states as opposed rotation matrices apply.

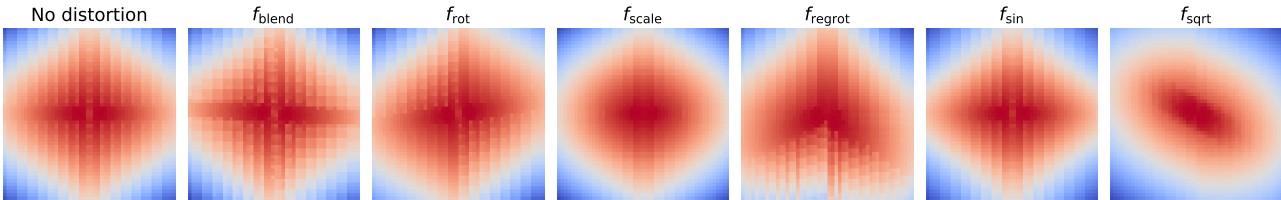

*Figure 9.* Value functions $V^\pi(\cdot, W)$ of coordinate walk for a random but fixed context $W$ each.

## C. Additional details for structured logging policies

This section provides additional details on the coordinate walk policy $\pi_{\text{cw},l}$ introduced in Section 5.1.3 and some insights on optimal policies for active positioning tasks. Figure 10 illustrates how the step size $l$ impacts the expertness of the coordinate walk policy in terms of the average number of steps to reach $s_W$. As designed, smaller step sizes lead to more expert behavior.

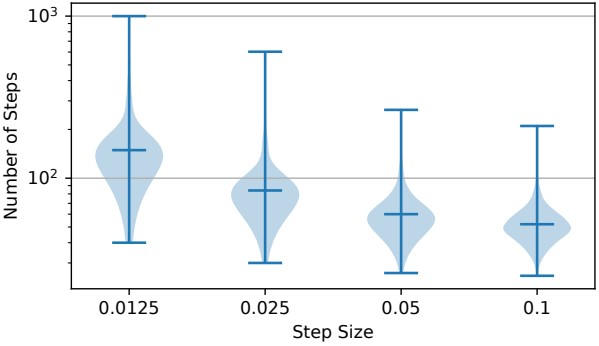

*Figure 10.* Expertness of $\pi_{\text{cw},l}$.

The coordinate walk policy interacts quite differently with the various movement distortions. Figure 11 shows example trajectories of the coordinate policy for different movement distortions. There, we also compare to a direct policy $\pi_{\text{direct}}$ that always takes the largest possible step $\text{clip}_\lambda(s_W - s)$ towards the goal.

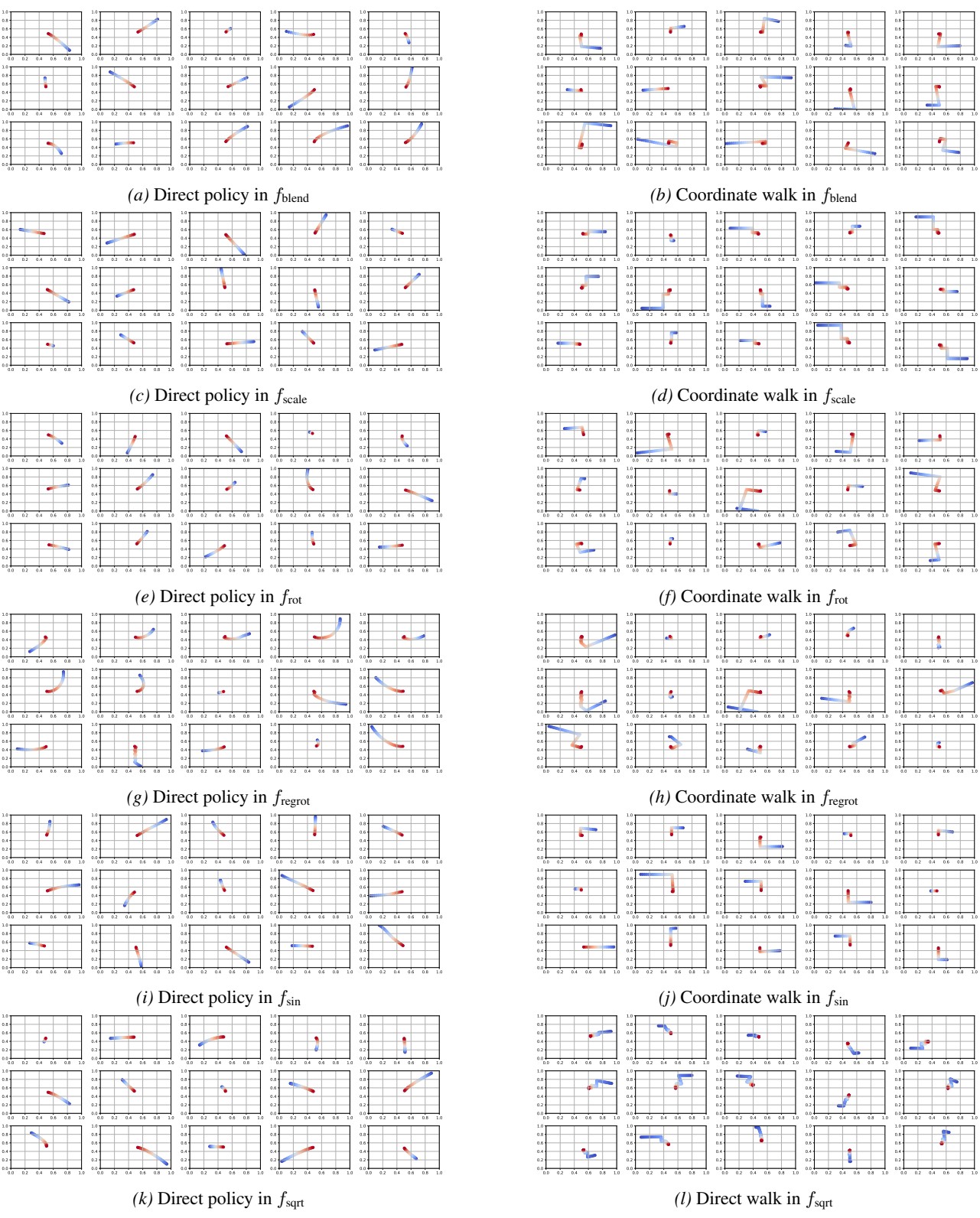

*(a)* Direct policy in $f_{\text{blend}}$

*(b)* Coordinate walk in $f_{\text{blend}}$

*(c)* Direct policy in $f_{\text{scale}}$

*(d)* Coordinate walk in $f_{\text{scale}}$

*(e)* Direct policy in $f_{\text{rot}}$

*(f)* Coordinate walk in $f_{\text{rot}}$

*(g)* Direct policy in $f_{\text{regrot}}$

*(h)* Coordinate walk in $f_{\text{regrot}}$

*(i)* Direct policy in $f_{\text{sin}}$

*(j)* Coordinate walk in $f_{\text{sin}}$

*(k)* Direct policy in $f_{\text{sqrt}}$

*(l)* Direct walk in $f_{\text{sqrt}}$

*Figure 11.* Trajectories of direct policy and coordinate walk in different movement dynamics.

Under mild distortions and additional assumptions on the distribution of $s_W$, the direct policy is the optimal policy

the optimal behavior under mild distortion and additional assumptions on the distribution of $s_W$. First, in case of full observability, the optimal policy is as follows:

**Proposition C.1.** *Under full observability, i.e., $O(s, W) = (s, W)$, the optimal policy $\pi_\lambda^*$ is given by*

$$\pi_\lambda^*(s, W) = \arg \min_{\|a\| \leq \lambda} \|f(s, a, W) - s_W\|.$$

*Proof.* First, we define the state-action value functions $Q^\pi(s, a, W)$ and $Q^\pi(s, a)$ similarly to the value functions $V^\pi(s, W)$ and $V^\pi(s)$ from Section 2. Clearly, the policy $\pi_\lambda^*$ is the policy yielding the maximal expected reward in each step. This is due to the fact as it gets closest to the terminal state $s_W$ and the reward depends only on the distance to $s_W$. Thus

$$\max_a Q^\pi(s, a, W) \leq \max_a Q^{\pi_\lambda^*}(s, a, W)$$

for any state $(s, W)$ and the same holds for the expected values over $W \sim \mathcal{W}$, i.e., $\max_a Q^\pi(s, a) \leq \max_a Q^{\pi_\lambda^*}(s, a)$. □

Clearly, the policy $\pi_\lambda^*$ from Proposition C.1 is not applicable in practice as neither the context $W$ is observed nor the movement dynamics $f$ is explicitly known which is needed to solve the minimization problem in each step. In case only $s$ is observed as in $\mathcal{O}_{\text{PO}}$, the best action a policy can take is the one where the expected distance to the terminal state over all contexts $W$ is minimized, that is:

$$\pi_\lambda^*(s) = \arg \min_{\|a\| \leq \lambda} \mathbb{E}_{W \sim \mathcal{W}}[\|f(s, a, W) - s_W\|].$$

Still, without further assumptions on $f$, $s_W$, and $\mathcal{W}$, computing $\pi_\lambda^*(s)$ is intractable. However, assuming the expected value of $s_W$ exists and is available and that the placement error does not depend on the state, i.e., $f(s, a, w) = s + g(a, W)$, the optimal is explicitly given as follows:

**Proposition C.2.** *Let $f(s, a, W) = s + g(a, W)$ with $\mathbb{E}_{W \sim \mathcal{W}}[g(a, W)] = a$ and assume that $\mathbb{E}_{W \sim \mathcal{W}}[s_W] = s^*$. Then the optimal policy is $\pi_\lambda^*(s) = \text{clip}_\lambda(s^* - s)$.*

*Proof.* We have

$$\pi_\lambda^*(s) = \arg \min_{\|a\| \leq \lambda} \mathbb{E}_{W \sim \mathcal{W}}[\|f(s, a, W) - s_W\|]$$
$$= \arg \min_{\|a\| \leq \lambda} \mathbb{E}_{W \sim \mathcal{W}}[\|s + g(a, W) - s_W\|]$$
$$= \arg \min_{\|a\| \leq \lambda} [\|s + a - s^*\|]$$
$$= \text{clip}_\lambda(s^* - s)$$

□

# D. Additional experiments in Fetch-environment

In extension to the reach experiments in Section 5 where the positional differences are directly observed, we provide in this section a proof of principle that shortcut augmentations can also benefit offline RL methods in more involved robotic environments. To this end, we consider two scenarios based on the Fetch environment (Plappert et al., 2018). In the first scenario, we study a reaching task in which the robotic arm must reach a target position in 3D space. The observation is an image of the scene. We collect 100 trajectories using the coordinate walk policy described in Section 5.1.3.

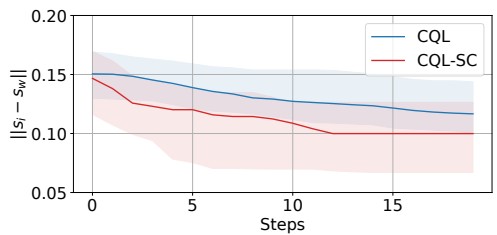

*(a)* Image-based reaching in $d = 3$ with 100 trajectories

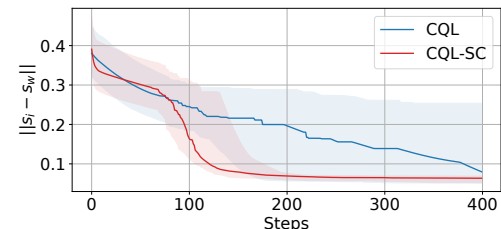

*(b)* Position based pick-and-place in $d = 3$ with 1000 trajectories

*Figure 12.* Experiments in the Fetch environment.

.

In the second scenario, we consider a variant of the pick-and-place task where the robotic arm must move an object from a random initial position to a random target position. We focus solely on the positioning, i.e., the object does not need to be grasped, only touched, assuming perfect gripper control. The policy used here performs two consecutive coordinate walks: one to reach the object and one to reach the target position. The observations are given by the distances from the gripper to the object and from the gripper to the target where the first distance is zeroed once solves the touching task. In this setting, we collect 1000 trajectories. On the collected datasets, we train CQL both with and without shortcuts, and the results are reported in Figure 12.

## E. Details for Experimental Results

### E.1. Hyperparameters of learning algorithms

| Parameter | Value |
|---|---|
| actor learning rate | $10^{-3}$ |
| critic learning rate | $10^{-3}$ |
| conservative weight | 5.0 |
| $\alpha$-threshold | 10.0 |
| batch size | 500 |
| $\gamma$ | 0.99 |
| $\tau$ | 0.005 |

*Table 1.* Parameter for CQL trained on collected datasets.

| Parameter | Value |
|---|---|
| actor learning rate | $10^{-3}$ |
| critic learning rate | $10^{-3}$ |
| conservative weight | 5.0 |
| $\alpha$-threshold | 10.0 |
| batch size | 500 |
| $\gamma$ | 0.99 |
| $\tau$ | 0.005 |

*Table 2.* Parameter for CQL trained as LIFT augmentor.

| Parameter | Value |
|---|---|
| actor learning rate | $10^{-3}$ |
| critic learning rate | $10^{-3}$ |
| batch size | 256 |
| n updates per step | 5 |
| n critics | 2 |
| $\gamma$ | 0.99 |
| $\tau$ | 0.005 |

*Table 3.* Parameter for SAC.

### E.2. Hyperparameter study of LIFT

In this section, we study effects of the different hyperparameters of the shortcut computation (Algorithm 1) and LIFT (Algorithm 2). First, we study the effect of the number of augmentations per trajectory $n$ and the probability of applying an augmentation $p$. The results are shown in Figure 13. One can see that as few as 20 augmentations per trajectory are sufficient to achieve a substantial improvement in performance, provided that the augmentation probability is not too low. Notably, higher probabilities correspond to augmentations being applied earlier in the trajectory. This suggests that augmentations at the beginning of a trajectory are more beneficial than those applied later.

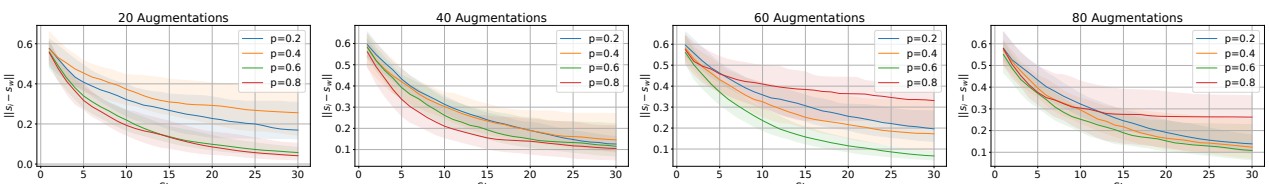

*Figure 13.* Experiments in $f_{\text{blend}}$ with step size 0.025 and different probabilities $p$ of applying augmentations and different maximal number of augmentations per trajectory

Next, we analyse the effect of the sampling scheme of shortcuts along a trajectory. Here, we denote the sampling mechanism described in Algorithm 1 as *weighted*. Another way to sample shortcuts from the set $S$ computed in Algorithm 1 is to use a distribution that is proportional to the inverse distance to the optimum, i.e. $p(i) \sim \frac{1}{\|s_i - s_W\|}$ or to sample uniformly from $S$. Instead of sampling, one can also just use the shortcut residing within the action space that leads to the point of highest reward within the trajectory called *best*. The results are shown in Figure 14 for $n = 20$ augmentations per trajectory and $p = 0.4$ showing that in the environments we consider, the sampling strategy does not have a significant effect on the performance.

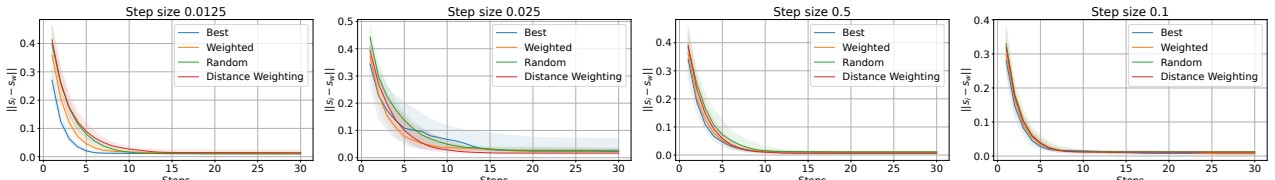

*Figure 14.* Experiments in $f_{\mathrm{blend}}$ with different step size and different sampling strategies.

### E.3. Comparison of LIFT and SAC

Table 4 summarizes settings in which LIFT-SC achieves a smaller distance to the optimum than the SAC baseline after 30 interaction steps in environment $\mathcal{O}_{\mathrm{PO}}$ with dimensionality $d = 5$, across different step sizes of the logging policy and movement distortions. Figures 19–24 provide a complete comparison of all methods over the first 30 steps, showing the median distance to the target across multiple runs.

| $\pi_{\mathrm{cw},l}$ | .0125 | .025 | .05 | .1 |
|---|---|---|---|---|
| $f_{\mathrm{blend}}$ | ● | ● | ● | ● |
| $f_{\mathrm{scale}}$ | ● | ● | ● | ● |
| $f_{\mathrm{rot}}$ | ● | ● | ● | ● |
| $f_{\mathrm{regrot}}$ | ● | | | |
| $f_{\mathrm{sin}}$ | ● | ● | ● | ● |
| $f_{\mathrm{sqrt}}$ | | ● | ● | ● |

*Table 4.* Cases where LIFT-SC outperforms SAC baseline in $\mathcal{O}_{\mathrm{PO}}$, $d = 5$.

### E.4. Ablation on structure of logging policy

In this section, we analyse the effect of absence of structure in the logging policy on the performance of the shortcut augmentation by injecting noise into the $\pi_{\mathrm{cw},l}$. Specifically, we used $\mathcal{O}_{\mathrm{PO}}$ under three different dynamics. At each step of the coordinate-walk logging policy, we added Gaussian noise to the action and considered a range of noise levels, from $\lambda = 0$ (the original coordinate walk) up to $\lambda = 2$, where the behavior is close to a random walk and little of the original coordinate structure remains visible (see Figure 15). We then train and evaluate three CQL models with and without shortcut augmentation respectively on datasets generated by these noisy-variant of $\pi_{\mathrm{cw},l}$. The results are in shown in Figure 16: Across all tested scenarios, shortcut augmentation consistently yields substantially better policies, suggesting that the method is not limited to highly structured logging policies.

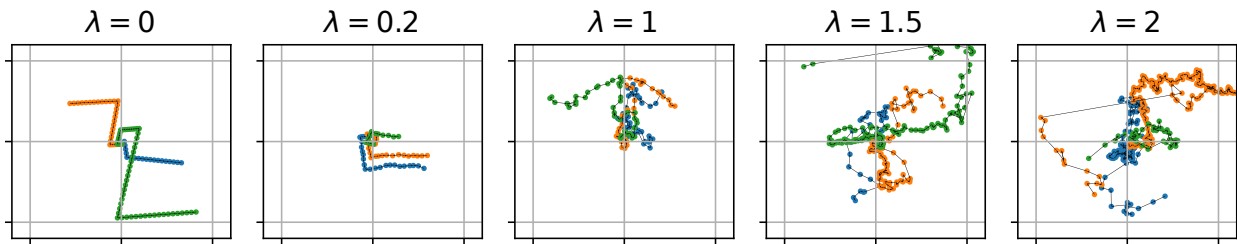

*Figure 15.* Comparison of different logging policies in $f_{\text{blend}}$ with $d = 5$ and step size $0.05$.

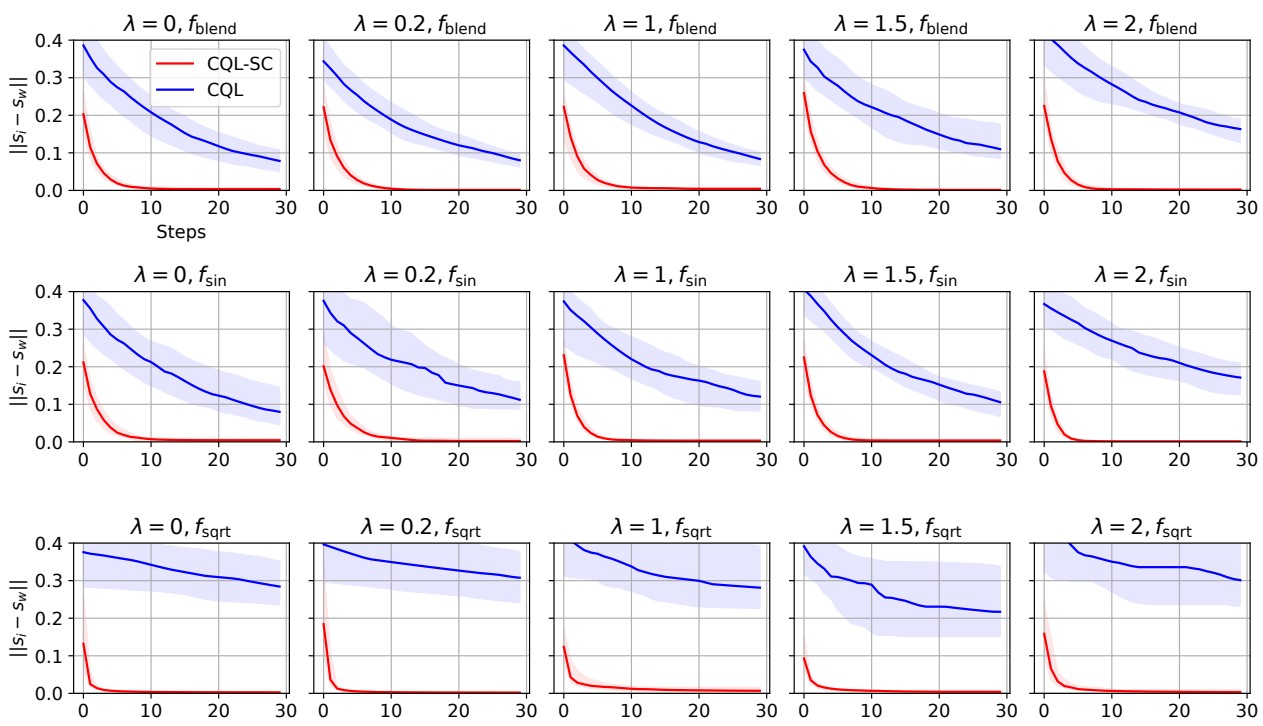

*Figure 16.* Comparison of noisy $\pi_{\text{cw},l}$ with different noise levels $\lambda$ for different movement distortions.

### E.5. Analysis of the Influence of $C$

In this section, we study the influence of the hyperparameter $C$ during shortcut computation (Algorithm 1). Higher values of $C$ lead to more restrictive shortcut selection.

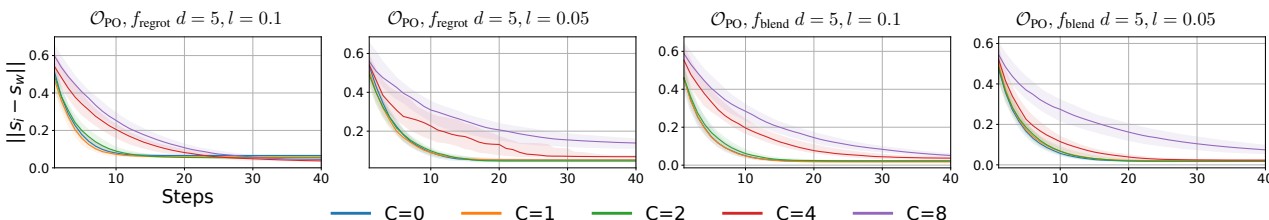

*Figure 17.* Comparisons of our methods for selected scenarios.

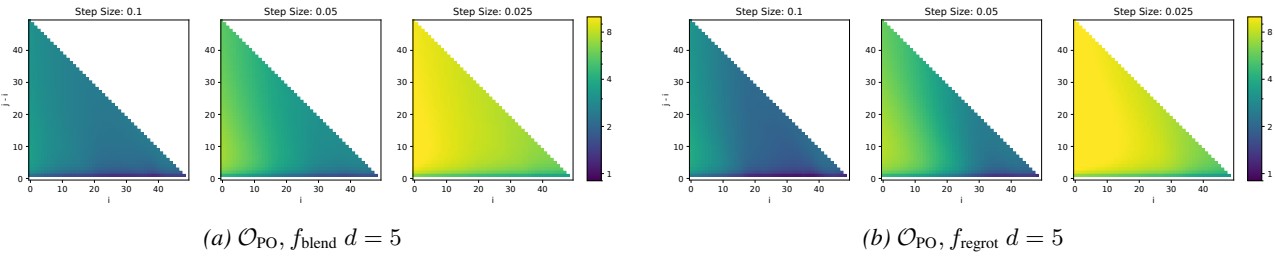

*(a)* $\mathcal{O}_{\text{PO}}$, $f_{\text{blend}}$ $d = 5$    *(b)* $\mathcal{O}_{\text{PO}}$, $f_{\text{regrot}}$ $d = 5$

*Figure 18.* Dependence which values of $C$ give valid shortcut from $i$ (x-axis) to $j - i$ (y-axis), averaged over 500 episodes of $\mathcal{O}_{\text{PO}}$.

## E.6. Additional visualization

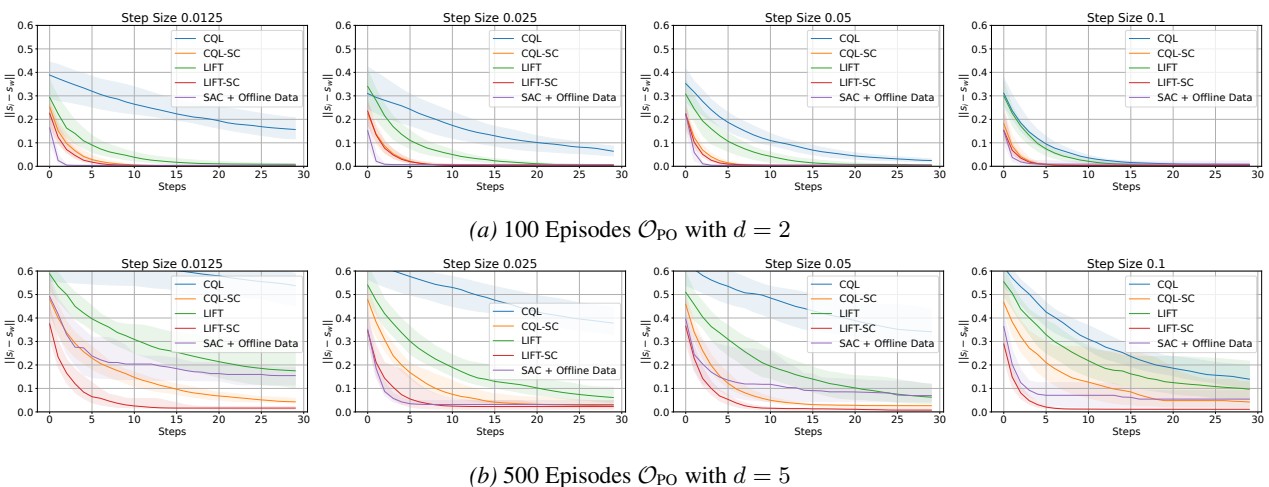

*(a)* 100 Episodes $\mathcal{O}_{\text{PO}}$ with $d = 2$

*(b)* 500 Episodes $\mathcal{O}_{\text{PO}}$ with $d = 5$

*Figure 19.* Experiments in $f_{\text{blend}}$.

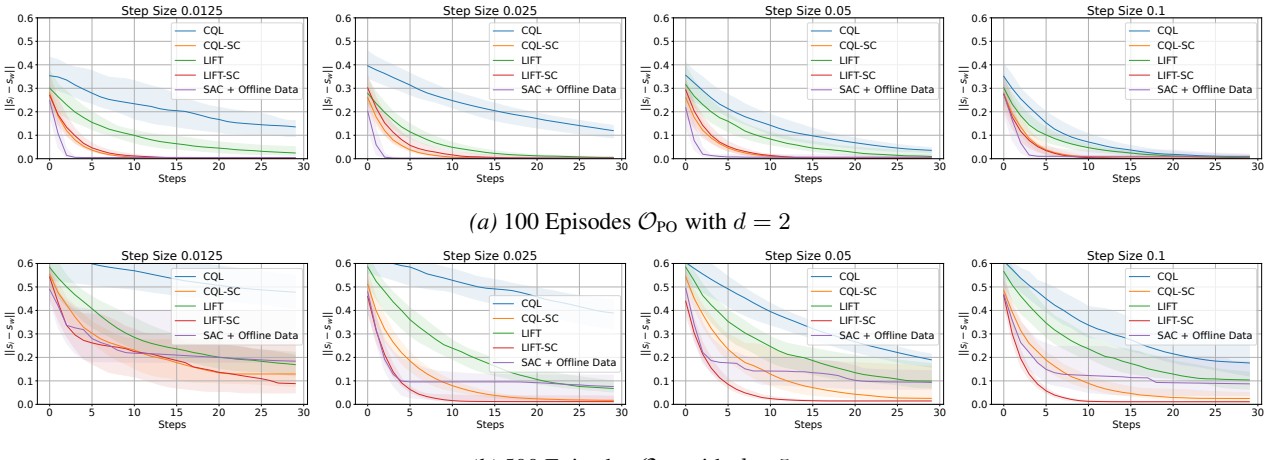

*(a)* 100 Episodes $\mathcal{O}_{\text{PO}}$ with $d = 2$

*(b)* 500 Episodes $\mathcal{O}_{\text{PO}}$ with $d = 5$

*Figure 20.* Experiments in $f_{\text{scale}}$.

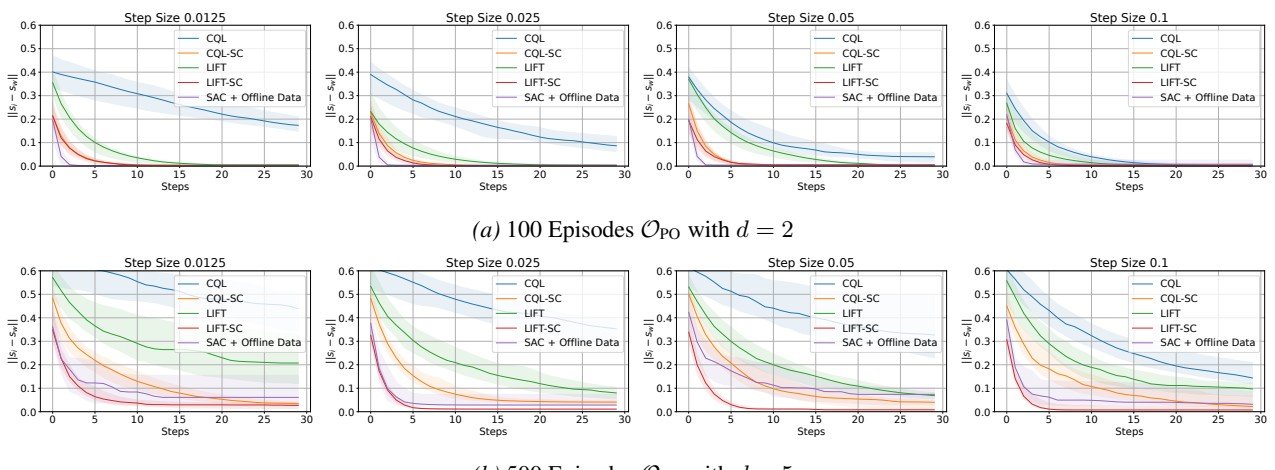

*(a)* 100 Episodes $\mathcal{O}_{PO}$ with $d = 2$

*(b)* 500 Episodes $\mathcal{O}_{PO}$ with $d = 5$

*Figure 21.* Experiments in $f_{rot}$.

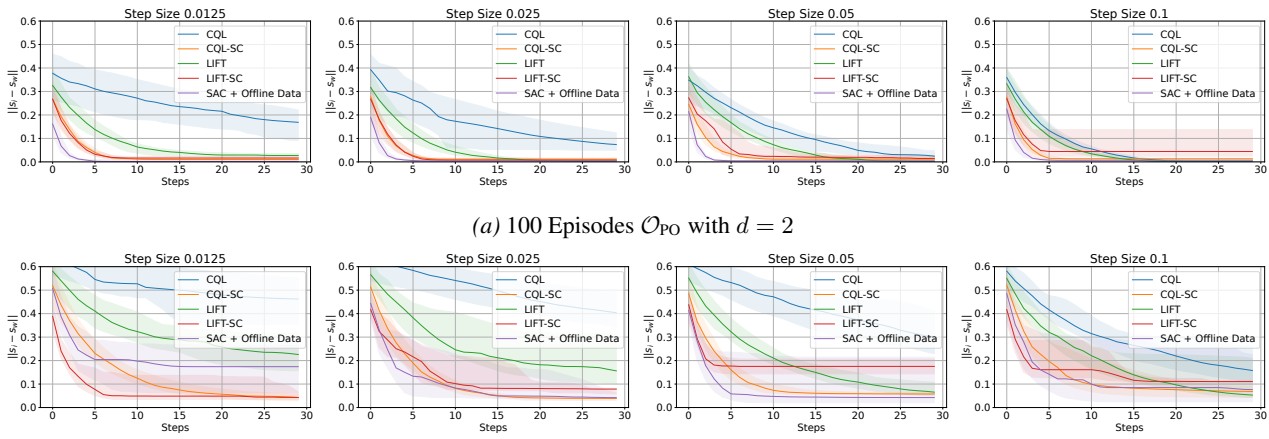

*(a)* 100 Episodes $\mathcal{O}_{PO}$ with $d = 2$

*(b)* 500 Episodes $\mathcal{O}_{PO}$ with $d = 5$

*Figure 22.* Experiments in $f_{regrot}$.

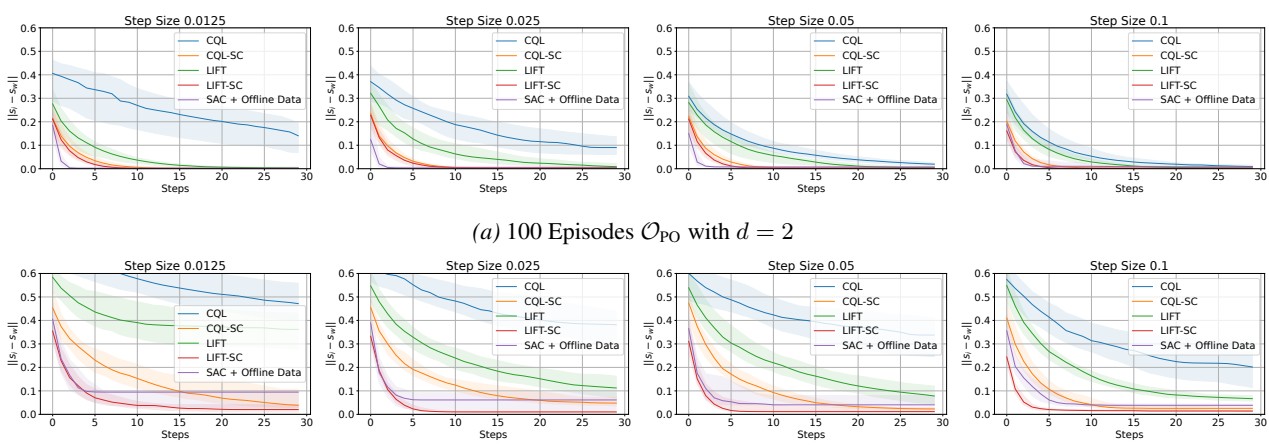

*(a)* 100 Episodes $\mathcal{O}_{PO}$ with $d = 2$

*(b)* 500 Episodes $\mathcal{O}_{PO}$ with $d = 5$

*Figure 23.* Experiments in $f_{sin}$.

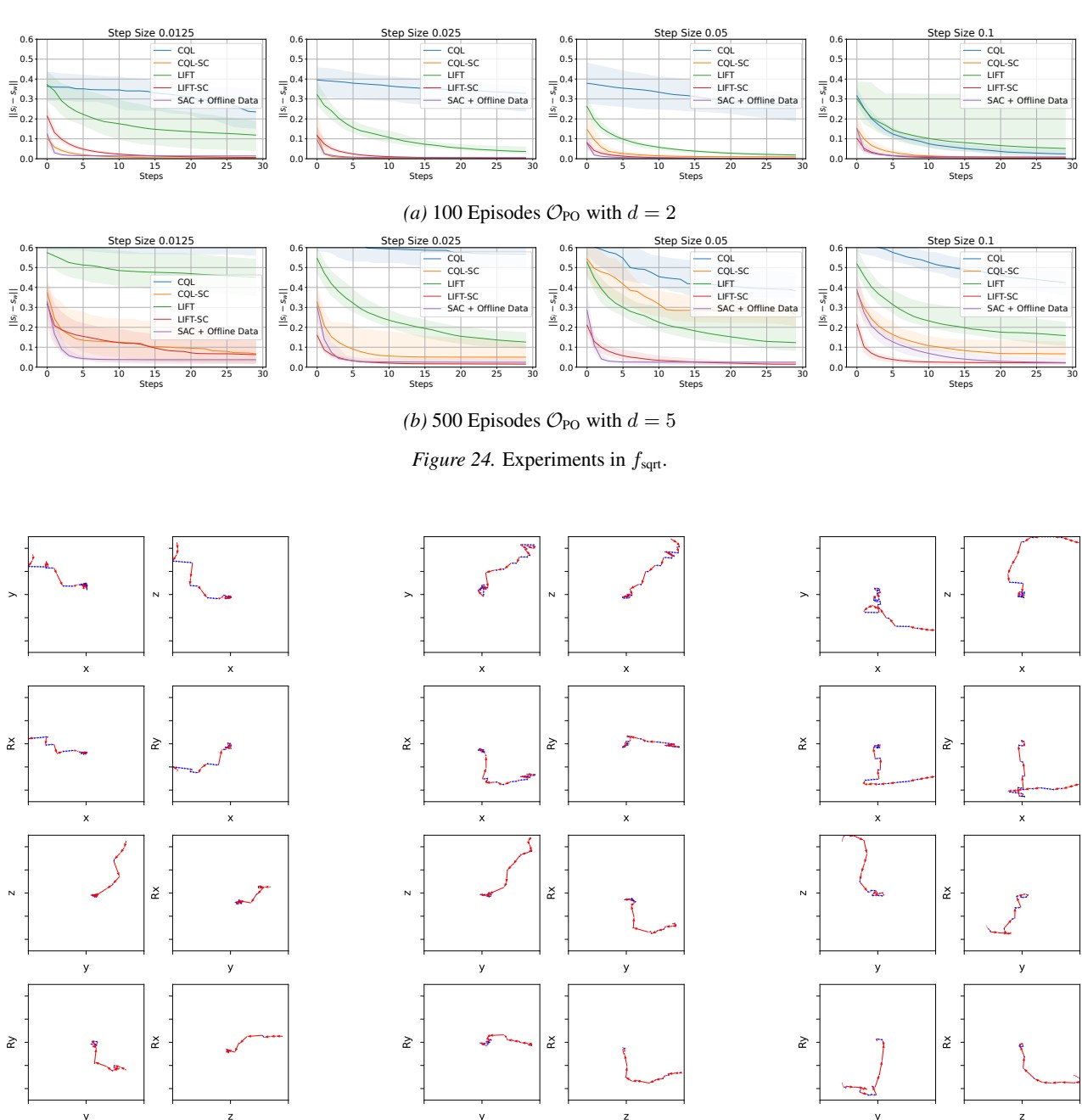

*(a)* 100 Episodes $\mathcal{O}_{\text{PO}}$ with $d = 2$

*(b)* 500 Episodes $\mathcal{O}_{\text{PO}}$ with $d = 5$

*Figure 24.* Experiments in $f_{\text{sqrt}}$.

*(a)*    *(b)*    *(c)*

*Figure 25.* Augmented trajectories generated by LIFT for $\mathcal{O}_{\text{LP}}$ in 5 dimensional hidden position space: Actions coming from the augmentor in red and actions from the logging policy in blue.

