# OpenReview forum: "Trajectory-Level Data Augmentation for Offline Reinforcement Learning"
_ICML.cc/2026/Conference — ICML 2026 regular_

### Official Review · Reviewer_Wkj6 · 2026-02-25

**Soundness:** 4
**Presentation:** 4
**Significance:** 3
**Originality:** 4
**Overall Recommendation:** 5
**Confidence:** 4

**Summary:**

The paper "Trajectory-Level Data Augmentation for Offline Reinforcement Learning" proposes a trajectory-level augmentation framework that improves offline RL datasets by selectively inserting value-improving shortcut actions into logging-policy rollouts.

The paper is well written, and the method and results appear solid. The authors tackle a challenging and significant problem in RL of obtaining better data from what are generally finely-crafted demonstrators (logging policies).

**Compliance With Llm Reviewing Policy:**

Affirmed.

**Key Questions For Authors:**

- The main question I have is how this would compare to learning a model. I am interested to know how the proposed approach compares with a simple model-based baseline or what a short-horizon rollout comparison would look like (e.g. fitting local dynamics and generating short, synthetic rollouts to a goal under the same data budget). Such a result would clarify the benefit of logging policy augmentation versus model rollouts. I do not view this comparison as necessary for acceptance. However, even a lightweight model-based baseline would sharpen the paper, and the method in general would I think benefit from a deeper comparison or ablation.

**Limitations:**

yes

**Strengths And Weaknesses:**

- I am not an expert in active positioning, but I find the core idea sound and the results convincing. Despite this, one of the main weaknesses of the current paper is a lack of validation on more realistic simulators or real-world platforms, as noted by the authors in the discussion. Nevertheless, the existing results present convincing evidence that the proposed approach would apply to other regimes.

- One thing I missed in the current manuscript are how much the augmentations actually reduce the trajectories that are collected. For instance, how many shortcuts are applied, and what is the effective trajectory length reduction? This information would provide more interpretability and better intuition about the reason why we see performance gains using LIFT. Perhaps this is buried somewhere, but would be useful to know.

---

> ### Author Rebuttal · Authors · 2026-03-30
>
> We thank the reviewer for the very positive assessment of our work and for the insightful questions.
> Below are our responses to the specific questions raised.
>
> ### Trajectory lengths
>
> Section E.2 already includes a sensitivity analysis for both the augmentation probability $p$ and the
> maximum number of augmentations per trajectory. Across our experiments, the best performance is
> achieved with $p=0.6$ and a maximum of 20 augmentations per trajectory. In this regime, the augmentation budget is
> typically fully utilized, indicating that the augmentor consistently finds useful interventions.
>
> Regarding trajectory length: This depends strongly on the step size of the logging policy. For
> instance, with step size 0.0125, the average trajectory length is reduced from roughly 90 to 70
> steps when using the augmentor. For larger step sizes, the logging trajectories are generally
> longer. That said, our objective is not to shorten trajectories for its own sake, but to generate
> data that is more useful for downstream offline RL. In our view, the main benefit of the augmentor
> is therefore improved data quality rather than trajectory compression alone. We agree, however, that
> an explicit analysis of trajectory-length reduction would be valuable, and we will include this in
> the final version.
>
> ### Model-based methods
>
> The comparison to model-based methods is an important point. Our current study focuses on model-free
> methods, but we have conducted initial experiments with offline model-based baselines. In
> particular, for $O_{\mathrm{LP}}$, we were not able to obtain a competitive policy with an offline
> model-based approach, even without incorporating LIFT.
>
> More broadly, our preliminary evidence suggests that applying offline model-based methods in this
> setting is not straightforward. The highly structured data induced by the logging policy appears to
> make dynamics learning particularly sensitive to overfitting, which can in turn lead to weak
> generalization and poor downstream control. A fair evaluation therefore requires careful model
> design, tuning, and regularization, as well as a systematic study of how to mitigate this
> overfitting effect.
>
> For this reason, we do not want to overstate our current observations or draw strong conclusions
> without a much more careful comparison. At the same time, we believe LIFT is highly relevant to the
> model-based setting. A central challenge for offline model-based RL is that a model trained
> primarily on small, local actions may extrapolate poorly to larger movements. Shortcut augmentation
> directly addresses this issue by enriching the data with more informative transitions. We therefore
> believe that model-based methods could benefit substantially from LIFT, and from shortcut
> augmentation more generally. An in-depth comparison with model-based methods is currently part of our ongoing
> research, but it is beyond the scope of the present paper.

---

> > ### Author Rebuttal · Reviewer_Wkj6 · 2026-04-02
> >
> > The rebuttal addresses the trajectory-length question well, and I suggest adding a light empirical comparison to model-based baselines in the future to better clarify the benefit of augmentation over short-horizon rollouts.

---

### Official Review · Reviewer_Hm15 · 2026-03-05

**Soundness:** 2
**Presentation:** 3
**Significance:** 2
**Originality:** 3
**Overall Recommendation:** 3
**Confidence:** 4

**Summary:**

The paper introduces LIFT, a trajectory-level data augmentation framework for offline RL in active positioning tasks. It addresses the "hand-off problem" that occurs when exploratory actions disrupt the internal state of deterministic logging policies. The authors propose identifying geometric "shortcuts" in offline trajectories, providing theoretical bounds using Linear Placement Errors and f-contractions. These shortcuts are used to train an augmentor that skips redundant sub-trajectories. LIFT is evaluated on synthetic non-linear distortion environments and a simplified fetch robotic arm task, demonstrating improvements over other baselines.

**Compliance With Llm Reviewing Policy:**

Affirmed.

**Final Justification:**

I have read the authors' final clarifications, as well as the other reviews. I appreciate the authors taking the time to explain their pragmatic choice regarding the logging policy’s access to $s_W$.

I understand their argument: the logging policy acts as a stand-in for a real-world heuristic that can successfully infer the target from observations. However, my core concern remains. By granting the data collecting policy perfect, ground truth access to the target state, the resulting offline dataset is completely lacking of the perceptual noise, delays, and so on that a real world heuristic would inevitably suffer from when dealing with raw, ambiguous observations (like the lens images). This artificially isolates the control problem from the perception problem during data collection, resulting in trajectory geometries that are much cleaner and more predictable than reality. This setup inherently flatters a geometric shortcutting algorithm.

Furthermore, I stand by my concern regarding the physical abstractions. Modifying the "fetch" environment to "touch" rather than grasp sidesteps the discontinuous contact dynamics that make real-world robotic manipulation difficult, and those are exactly the dynamics that would severely stress-test their assumptions.

While I agree with the other reviewers that the theoretical framing is elegant and the problem is practically motivated, the empirical validation relies on artificially pristine data collection and oversimplified physical interactions. I don't believe the current experiments robustly validate the theory for the messy real-world systems it targets. Therefore, I respectfully maintain my original score of 3.

**Key Questions For Authors:**

1. Please address the questions in the weaknesses section above.
2. Seems like algorithm 1 relies on the hyperparameter $C$ to threshold valid shortcuts, yet there is no sensitivity analysis for $C$ in the appendix. How sensitive is the performance of LIFT to the choice of $C$? Could a practitioner supposed to safely tune this value without prior knowledge of the environment's distortion function?"
3. The evaluations are limited to $d=2$ and $d=5$ action spaces. Given your motivation of real-world robotic manipulation and complex optical alignments, how does the framework scale to higher dimensions? Does the probability of finding viable, safe shortcuts drop significantly as action dimensionality increases and movement distortions compound?

**Limitations:**

yes

**Strengths And Weaknesses:**

## Strengths
* The problem is well motivated, and addresses a known bottleneck in industrial reinforcement learning. The "hand-off problem" with stateful, deterministic logging policies seems like it could be a real problem for real-world control systems, and looking for geometric shortcuts in offline trajectories is a creative way to bypass it.
* The mathematical formalization is sound. Using Linear Placement-Errors (LPE) and f-contractions to bound movement errors provides a rigorous, logical foundation for defining when and why a shortcut is safe to take.

## Weaknesses
* The framework feels heavily overfit to active positioning tasks that use rigidly structured, axis-aligned logging policies (like your synthetic coordinate walk). How does LIFT handle more complex or black-box procedural policies? If a practitioner cannot cleanly reset the internal state of their policy after an augmented action, doesn't the system just fail or require a full episode restart?
* You motivate the framework as a POMDP where the target context $W$ and the exact target position $s_W$ cannot be observed directly. However, your synthetic coordinate walk logging policy explicitly requires "access to the target position $s_W$" (section 5.1.3) to compute its path. Wouldn’t evaluating an offline RL method on data generated by an expert that utilizes privileged, unobservable information fundamentally skew the difficulty of the task?
* The guarantees lean hard on LPE and f-contractions, but you note yourselves that simple non-linearities (like the $f_{sqrt}$ distortion) break the LPE assumption. When these assumptions inevitably fail in messy real-world systems, what would prevent the agent from taking a dangerously wrong shortcut?
* The paper is heavily motivated by real-world industrial complexities, but the evaluations abstract most of that away. For example, modifying the fetch environment so the arm only has to "touch" the object rather than grasp it ignores the discontinuous contact dynamics that might be a much harder robotic task. Did you consider how realistic physical interactions like grasping might completely break your shortcut assumptions?
* In Algorithm 1, you pair the shortcut action $\hat{a}$ with the final observation $o_j$. You acknowledge that the actual state reached $s'_j$ might differ from $s_j$ due to movement distortions, but assume this gap is small. However, for sensitive visual tasks, like your lens alignment where images are highly sensitive to microscopic misalignments (written in line 318, right column), this could create invalid transitions in the replay buffer, right? Could this transition error be the reason why CQL-SC fails and has large variance on the O_LP image task in Figure 6?

---

> ### Author Rebuttal · Authors · 2026-03-30
>
> We thank the reviewer for the detailed review and the insightful questions. Below is a point-by-point response to the concerns raised.
>
> - Our theoretical analysis is intentionally scoped to setting where clean resets of logging policies are available, as in the real-world problems our work is motivated by. We agree that extending LIFT to policies with non-resettable internal state can be an interesting open direction, although handling such policies would likely require first a precise mathematical formalization of resetability (alone what "cleanly" means) and hand-off behavior.
>
> - We agree that the coordinate-walk uses the target position $s_W$ in order to construct a structured path towards it. This is a deliberate modeling choice. Our goal is to study a practically relevant regime in which a reliable procedural policy reaches the target but does so in a highly inefficient and repetitive way. We do not believe this skew the difficulty of the task - if anything, it makes the offline RL problem challenging in a different way: because the logging policy almost always succeeds, the dataset consists of long, highly structured trajectories with limited diversity and strong procedural bias. As our experiments show, such data are difficult for standard offline RL methods despite the apparent success of the behavior policy.
> - We agree that the assumptions behind our guarantees are idealized and may fail in messy systems. Our intention is not to claim that all realistic distortions satisfy LPE globally, but rather to identify conditions under which shortcutting can be analyzed rigorously and to study empirically how sensitive our method is when these conditions are only approximately satisfied. Moreover, non-linearity alone is not what breaks the theory. In fact, several distortions are non-linear and still satisfy LPE. Thus, the relevant question is not whether the system is *simple* or *non-linear*, but whether action aggregation can produce placement errors that remain sufficiently controllable. What our experiments show is that shortcut augmentation remains useful beyond the exact theory.
> - We agree that our evaluation does not cover all of the physical complexity of discontinuous contact dynamics such as grasping. But this is a deliberate scope choice rather than an oversight. Our paper studies settings, where the main challenge is to move reliably and efficiently toward a target under distortion and partial observability and our theory is designed for exactly this regime. This is a meaningful setting in its own right: many industrial procedures decompose naturally into stages, where precise positioning is one challenge and contact-rich interaction is another. We want to emphasize that beyond the Fetch setup studied commonly, we also consider higher-dimensional robotic reaching problems using image-based observations only. These experiments are meant to test whether the shortcut idea remains useful under limited observability and structured but imperfect data, rather than under full contact-rich manipulation. We therefore agree with the reviewer’s broader point: realistic grasping dynamics could violate our shortcut assumptions, and we do not currently provide guarantees in that setting. A natural next step would be to apply LIFT to multi-stage manipulation pipelines, where shortcutting is used for the positioning phase and combined with a separate method for the contact-rich phase.
> - We noticed that the captions for LT and LP were accidentally swapped in Figure 6.
>   To clarify, the middle-bottom in Figure 6 shows $\mathcal{O}_{\mathrm{LP}}$ and the high variance
>   is not from CQL-SC. We apologize for the confusion, as well as for the poor color
>   mapping, and will improve both in the final version. We agree
>   that the approximation in Algorithm 1 is only justified when the induced gap is small. However,
>   since CQL-SC shows robust and strong performance on both visually sensitive tasks,
>   we believe our approximation is reasonable.
>
> ## Answers to questions
>
> - Although $C$ plays an important role in our theoretical analysis, we did not find shortcut computation to be highly sensitive to its precise value in our empirical study. We agree, however, that documenting this sensitivity would help clarify the role of C for readers. We will therefore include a dedicated sensitivity analysis in the final version.
> - In the settings we study, the framework appears to scale favorably with increasing action dimensionality. As noted in the discussion, the largest gains occur in more complex, higher-dimensional settings. Our intuition is that higher-dimensional structured policies generate longer, more redundant trajectories, creating more opportunity for beneficial shortcuts. That said, our experiments only cover moderate dimensions, so scaling to very high-dimensional systems remains subject of our current research.

---

> > ### Author Rebuttal · Reviewer_Hm15 · 2026-04-03
> >
> > Thanks to the authors for the rebuttal.
> >
> > However, I selected option (c) because the authors essentially agreed with my main structural critiques: The reliance on cleanly resettable states, the abstraction of contact dynamics, and most importantly, the fact that the logging policy uses privileged, unobservable information.
> >
> > While the authors argue these are deliberate scope choices, it confirms my original concern: the framework is highly overfit to a very specific, somewhat artificial setup. Feeding an offline RL algorithm data from an expert that is "cheating" by using unobservable variables fundamentally changes the nature of the POMDP task in a way that limits the broader applicability of the findings.
> >
> > I will maintain my original score.

---

> > > ### Author Response · Authors · 2026-04-05
> > >
> > > We thank the reviewer for the feedback. We believe the concern about “privileged information” arises from a misunderstanding of our setup, and we would like to take this opportunity to clarify this point.
> > >
> > > In our synthetic environments, the goal is to move from a randomized initial position $s$ to a
> > > target position $s_W$ under an unknown distortion $s' = f(s, W)$, where $W$ is a randomized
> > > context variable. The policy observes $O(s, W)$ and must act based on this
> > > observation; the full setup is defined in Section 2. Across all scenarios we study, successful
> > > control requires that relevant displacement information — essentially $s - s_W$ — is inferable from
> > > the observation, since otherwise the task is not solvable. Even when $s - s_W$ is inferable, however,
> > > the task may still be unsolvable without any information about the movement distortion. This
> > > requirement is discussed more formally in Appendix C and instantiated concretely in Section 5.1.2
> > > for the various observation settings we study (e.g., if only the position $s$ is observed, one must
> > > use a constant target position).
> > >
> > > To generate reliable trajectories across these different observation settings and distortion
> > > regimes, we gave the logging policy direct access to $s - s_W$. Importantly, this does not make the
> > > task trivial, nor does the policy "cheat" — we simply assume the logging policy already has a reliable way to
> > > infer the relevant information from the observation. Since the movement dynamics are unknown to the
> > > policy, knowledge of $s - s_W$ alone is in general not sufficient to find a path to the target. Instead, inspired by
> > > real-world logging policies as described in Section 5.1.3, we constructed a structured logging
> > > policy that optimizes coordinate by coordinate. As a result, the logging policy can reach the target
> > > for the movement distortions we consider, but it does so inefficiently, including overshoots,
> > > detours, and movement in the wrong direction (see Fig. 11 for example trajectories).
> > >
> > > The reason we gave the logging policy access to $s - s_W$ is purely pragmatic, especially in settings
> > > with image observations: it allows us to freely combine movement distortions and observation
> > > settings while using the same structured logging policy, and to analyze the effects of different
> > > distortions and observation settings in a more controlled way.
> > >
> > > We hope this clarification helps the reviewer see our setting more precisely and reassess this point accordingly.

---

### Official Review · Reviewer_M8pa · 2026-03-08

**Soundness:** 3
**Presentation:** 1
**Significance:** 2
**Originality:** 3
**Overall Recommendation:** 4
**Confidence:** 3

**Summary:**

The paper studies how to improve the logging policy for downstream offline RL problems. The proposed method, LIFT, is a data augmentation approach designed specifically for active positioning problems, a particular class of POMDP in which rewards are given based on the distance between current and target state. LIFT trains an augmentor $a_\theta$ on the offline dataset, which is expected to assist the data collection process by taking actions with higher return. To achieve this, the offline dataset is augmented with "good transition", which correspond to shortcuts in the active positioning problem. The paper gives a detailed theoretical analysis on how to find shortcuts from the offline dataset, and applies it to the actual training process of $a_\theta$.

**Compliance With Llm Reviewing Policy:**

Affirmed.

**Final Justification:**

Most of my concerns are related to the writing of the paper. As long as the concepts are clearly explained in the revision, the paper is good. The rebuttal reinforced my prior assessment, so I will keep the positive score.

**Key Questions For Authors:**

I need some clarifications for Algorithm 2:
1. Is the returned dataset $D$ used as a fixed offline dataset to train downstream model?
2. Is the input $a_\theta$ trained or randomly initialized? If it is trained, on what data and how is it trained?
3. Since $a_\theta$ is expected to be better than $\pi_\beta$, why can't I sample the whole trajectory purely with $a_\theta$?

**Limitations:**

Yes.

**Strengths And Weaknesses:**

**Strength**
1. The paper makes a good combination of theory and practice. First, a comprehensive theoretical analysis is made on how to find shortcuts in the trajectory (Theorem 3.6). Then the theorem is applied to train an augmentor that is likely to augment the dataset with better trajectory.
2. Comparing to vanilla data augmentation, which randomly stitches trajectories in dataset, LIFT augments the dataset with trajectories

**Weakness**
1. LIFT only applies to action position problems, which is a restricted class of POMDP.
2. The title of the paper is misleading. The title gives a false feeling that the paper is about RL *training*: A *model* needs to be trained with a fixed dataset without interaction with the environment, and synthetic data is augmented to improve performance. In fact, the paper is about *data collection*: We have access to a logging policy and some pre-collected trajectories, and now we are going to sample more trajectories from the environment - How can we sample trajectories with higher quality? A better title can be, e.g., "Trajectory-Level Data Augmentation for Logging Policy Improvement".
3. The pseudo-code is confusing. In Algorithm 2, line 16, $a_\theta$ is trained with Algorithm 1, but Algorithm 1 is not a training algorithm.
4. The improvement of logging policy is not a strict offline setting. Since it is used for data collection, the environment is still accessible. Therefore, online RL approaches can also be used and should be compared with LIFT.
5. More related literature: Since the paper uses the idea of data augmentation, the paper should also cite prior data augmentation works in the general offline RL setting:
[1] Ghugare, Raj, et al. "Closing the Gap between TD Learning and Supervised Learning--A Generalisation Point of View." arXiv preprint arXiv:2401.11237 (2024).
[2] Zhou, Zhaoyi, et al. "Free from bellman completeness: Trajectory stitching via model-based return-conditioned supervised learning." arXiv preprint arXiv:2310.19308 (2023).

---

> ### Author Rebuttal · Authors · 2026-03-30
>
> We thank the reviewer for the positive feedback and helpful suggestions. We respond to each comment in detail below.
>
> ### Answer to Weakness
> - Although our empirical evaluation focuses on problems inspired from active positioning, our
> theoretical findings and methodology are applicable to a broader class of problems.
>
> - We appreciate the reviewer's feedback; however, we believe the title is appropriate because our augmentation is specifically designed to improve the final offline RL training phase. Furthermore, since the method can be applied post-hoc to existing static datasets (e.g., CQL-SC), our contribution extends beyond just the data collection process.
>
> - The reviewer is correct that Algorithm 1 is not a training algorithm itself, but rather a method to provide augmented data for a subsequent offline RL training. We will clarify this in the final version to ensure it is clear that the RL algorithm is trained *with the help of* Algorithm 1.
> -  We thank the reviewer for this remark. Since LIFT-SC involves environment interaction during data collection, we explicitly compared it against a online RL baseline (SAC). To ensure a fair comparison, SAC was initialized with the same offline data ('warm-start') and restricted to the exact same data budget (total number of environment interactions) as LIFT-SC.
>
> - We thank the reviewer for the pointer to more related literature. The work "Closing the Gap between TD Learning and Supervised Learning--A Generalisation Point of View." is closely related but focuses on the lack of combinatorial generalization (‘stitching’) in supervised RL methods, whereas our work addresses a complementary problem at the data level through trajectory-level augmentation. We will include a discussion of this work in our related work section.
>
> ### Answer to Key Questions
>
> - Yes, the resulting dataset is fixed and used for standard offline RL training.
>
> - While Algorithm 2 is executed, the number of trajectories in $D$ increases. Once a sufficient amount of data has been collected, the augmentor $a_\theta$ is trained on $D$ (details when this happen can be found on Page 8, Line 410-417 on left column)
>
> - Since $a_\theta$ is only trained on a limited amount of data, its actions are generally only occasionally better than those of the logging policy. The effects of increasing the number of actions in the policy are shown in Section E.2, Figure 13. There, an optimum was also determined at $p = 0.6$ with a maximum augmentation of $20$.

---

> > ### Author Rebuttal · Reviewer_M8pa · 2026-04-01
> >
> > Most of my concerns are related to the writing of the paper. As long as the concepts are clearly explained in the revision, the paper is good. I believe my initial understanding of the paper is correct, so I will keep the score.

---

> > > ### Author Response · Authors · 2026-04-02
> > >
> > > We sincerely appreciate your time and careful questions -  we are glad that our clarification has resolved your concerns! Thank you for acknowledging the contributions of our work and keeping your positive evaluation.

---

### Official Review · Reviewer_66yx · 2026-03-11

**Soundness:** 2
**Presentation:** 3
**Significance:** 3
**Originality:** 3
**Overall Recommendation:** 4
**Confidence:** 2

**Summary:**

This paper proposes a trajectory-level data augmentation framework LIFT, addresses the data quality dependency problem in offline reinforcement learning by proposing. It improves offline dataset quality by identifying and exploiting "shortcuts" within already-recorded trajectories at the data collection stage, thereby enhancing the learning performance of subsequent offline RL policies. The paper provides extensive and precise formal definitions for the theoretical concepts involved (e.g., Shortcut Augmentations), offers theoretical analysis of shortcut validity, and conducts comparative experiments against methods such as CQL, SAC, GTA, and DQL across various motion distortion functions and observation types. Both theoretical validation and empirical results demonstrate that shortcut augmentation consistently improves offline reinforcement learning performance on active localization tasks.

**Compliance With Llm Reviewing Policy:**

Affirmed.

**Final Justification:**

Thanks for the authors' responses, my concerns have been resolved. I will maintain my positive score.

**Key Questions For Authors:**

Please see Weaknesses.

**Limitations:**

Yes.

**Strengths And Weaknesses:**

## Strengths

**Soundness:** The paper establishes a clear and complete theoretical framework around Shortcut Augmentations, systematically defining core concepts such as LPE, f-contraction, and π-shortcut, and rigorously proving sufficient conditions for shortcut validity. Building on this foundation, the authors conduct comprehensive comparative experiments against representative methods including CQL, SAC, GTA, and DQL across multiple motion distortion functions and observation types, providing solid empirical support for the method's effectiveness.

**Presentation:** The paper is well-structured with rich figures and relatively detailed descriptions of experimental setups. Some core concepts (e.g., the intuitive motivation behind shortcuts) are explained accessibly, and extensive formal definitions are used to characterize the task setting for Shortcut Augmentations.

**Significance:** The data collection-stage augmentation strategy proposed in this paper addresses a practical pain point that has long been overlooked by the offline RL community and holds considerable industrial application value. However, the method's effectiveness strictly depends on the special structure of the active localization task; it has not been validated on general benchmarks (e.g., D4RL), which limits its influence on the broader ICML community.

**Originality:** The paper introduces action-shortcut-based data augmentation into the data collection stage and combines it with Q-function training. This approach is distinct from both pure offline augmentation and offline-to-online fine-tuning paradigms, offering a degree of novelty.


## Weaknesses

1. The theoretical proofs rely on strong assumptions: the logging policy must be distance-improving, the value function must satisfy a Lipschitz condition, the policy should ideally satisfy f-contraction, and whether a shortcut holds further depends on the LPE condition of the distortion function.

2. Without prior knowledge of the environment dynamics $f$ and the properties of the logging policy $\pi_\beta$, the precise value of the constant $C$ in the third paragraph of Section 4 cannot be determined from any prior, which limits the practical deployability of the method.

3. The logging policy in this paper is not a general off-the-shelf behavior policy, but rather a specially constructed coordinate walk with strong structural priors. Such a policy naturally generates reliable yet redundant trajectories, making it highly compatible with shortcut augmentation by design—and leaving the method's generalization to more general behavior policies yet to be verified.

---

> ### Author Rebuttal · Authors · 2026-03-30
>
> We thank the reviewer for the positive feedback to our work and help us strenghten it. We would like to clarify and respond to your concerns point by point.
>
> 1)  Although our theoretical results are based on strong assumptions, we believe there is
> significant potential to relax these conditions. Notably, our experiments indicate that the method
> still performs well even when some assumptions are only partially satisfied. For example, the
> logging policy is neither everywhere distance-improved nor a contraction for the given distortion globally. Moreover, certain distortions do not
> fully satisfy the LPE; yet the shortcut augmentation continues to yield improvements.
>
> 2) Theoretically, these parameters require specific values to compute the constants and guarantee
> performance. However, in practice, precise knowledge of these values is not always necessary for
> effective use, thereby highlighting a discrepancy between the strict theoretical requirements and
> practical applicability. As also other reviewers asked for the role of $C$ in practice, we will
> include an ablation study on $C$ to better understand its influence.
>
> 3)  Our theoretical investigation shows how to exploit geometric suboptimalities in any logging policy
> under some structured assumptions. The coordinate walk is a specific example that we use to
> illustrate these concepts, but the underlying principles are not limited to this particular
> strategy. At the same time, we intentionally chose the coordinate walk because it induces exactly the type of
> highly structured, repetitive behavior on which standard offline RL methods struggle, making it a
> meaningful stress test for our approach. Our results show that LIFT can effectively mitigate this
> difficulty. To address the reviewer’s concern more directly, we will add an ablation with a more randomized walk
> toward the target, in order to demonstrate that our method is not limited to structured policies.

---

> > ### Author Rebuttal · Reviewer_66yx · 2026-04-03
> >
> > Thanks  for the authors'  proactive response. Most of my questions have been addressed. However, since the authors’ answer to Q3 is limited to a theoretical explanation and lacks experimental results, I still have concerns about the generalization ability of the proposed method to general environments or general policies. In light of this, I will maintain my score and wait for the authors’ update.

---

> > > ### Author Response · Authors · 2026-04-03
> > >
> > > We thank the reviewer again for the thoughtful feedback and for helping us strengthen our results.
> > >
> > > We have just completed a first round of experiments with noise injected into the coordinate walk. At each step of the coordinate-walk logging policy, we added Gaussian noise to the action and considered a range of noise levels, from $\lambda=0$ (no noise) up to $\lambda=2$, where the behavior is close to a random walk and little of the original coordinate structure remains visible (see [this url](https://ibb.co/k2m8T6Hq) for example trajectories).
> > >
> > > Using the same hyperparameters as in the paper, we then trained and evaluated CQL with and without shortcut augmentation on datasets generated by these noisy logging policies under three different movement dynamics. Preliminary results are available [under this link](https://ibb.co/DFxtVh1). Across all tested scenarios, shortcut augmentation consistently yields substantially better policies, suggesting that our method is not limited to structured logging policies. We will include these experiments in the revised manuscript and hope the reviewer finds these additional results informative and helpful.

---

### Decision · Program_Chairs · 2026-04-30

**Decision:**

Accept (regular)

**Comment:**

**Summary:** This paper proposes LIFT, a trajectory-level data augmentation framework for offline reinforcement learning, motivated by active positioning problems. The main idea is to identify and exploit shortcut actions within logged trajectories in order to improve the quality of the offline dataset and thereby improve downstream offline RL performance. The paper provides a theoretical framework for when such shortcuts are valid, based on geometric properties such as linear placement errors and contraction-style assumptions, and validates the approach empirically on several active positioning tasks with different distortions, dimensions, and observation modalities. Overall, the paper addresses a practically relevant problem and presents a thoughtful combination of theory and experiments around a novel data-centric intervention for offline RL.


**Meta Review:** The reviewers were generally positive about the paper. Reviewer Wkj6 found the core idea sound and the empirical results convincing, and viewed the problem as important for improving data quality in structured offline RL settings. Reviewer 66yx also appreciated the clear theoretical framework and the systematic empirical comparisons, noting that both the formal analysis and experiments support the effectiveness of shortcut augmentation. Reviewer M8pa similarly viewed the paper as making a good combination of theory and practice, and after rebuttal indicated that the main concerns were largely about presentation and clarity rather than the core contribution. Taken together, the reviews support the view that the paper makes a meaningful contribution on an important applied RL problem.

There are, however, several points that should be improved in the final version. First, the authors should incorporate the rebuttal clarifications directly into the paper, especially regarding the precise scope of the method, the role of the logging policy, and the distinction between post-hoc offline augmentation and the data-collection setting used in LIFT-SC. Second, the paper should include the promised additions from rebuttal: the sensitivity analysis for the shortcut threshold parameter (C), the randomized-walk ablation addressing generalization beyond the coordinate-walk policy, and the clarified discussion of trajectory-length reduction. Finally, Reviewer Hm15 raised concerns about the realism of the empirical setup, especially the reliance on a structured logging policy with privileged access and the abstraction away from more contact-rich dynamics. While I believe the rebuttal satisfactorily clarified the intended scope and the role of the logging policy, the final version should still make these scope assumptions explicit and avoid overstating generality beyond the structured active-positioning regime studied here.

Overall, I recommend acceptance.